# Sex-determining genes distinctly regulate courtship capability and target preference via sexually dimorphic neurons

Kenichi Ishii[1†], Margot Wohl[1,2†], Andre DeSouza[1,2], Kenta Asahina[1,2]*

[1]Molecular Neurobiology Laboratory, Salk Institute for Biological Studies, La Jolla, United States; [2]Neuroscience Graduate Program, University of California, San Diego, San Diego, United States

**Abstract** For successful mating, a male animal must execute effective courtship behaviors toward a receptive target sex, which is female. Whether the courtship execution capability and upregulation of courtship toward females are specified through separable sex-determining genetic pathways remains uncharacterized. Here, we found that one of the two *Drosophila* sex-determining genes, *doublesex* (*dsx*), specifies a male-specific neuronal component that serves as an execution mechanism for courtship behavior, whereas *fruitless* (*fru*) is required for enhancement of courtship behavior toward females. The *dsx*-dependent courtship execution mechanism includes a specific subclass within a neuronal cluster that co-express *dsx* and *fru*. This cluster contains at least another subclass that is specified cooperatively by both *dsx* and *fru*. Although these neuronal populations can also promote aggressive behavior toward male flies, this capacity requires *fru*-dependent mechanisms. Our results uncover how sex-determining genes specify execution capability and female-specific enhancement of courtship behavior through separable yet cooperative neurogenetic mechanisms.

*For correspondence:
kasahina@salk.edu

†These authors contributed equally to this work

**Competing interests:** The authors declare that no competing interests exist.

## Introduction

Social behaviors, such as courtship and aggressive behaviors, are consequential for the fitness of many animal species. Successful male courtship behavior is vital for the propagation of offspring, whereas intra-specific aggressive behavior is often the key to ensure access to potential mates. These two behaviors therefore serve to directly and indirectly increase the reproductive success of males. Likely owing to the contribution to a common goal, e.g. reproduction, courtship and aggressive behaviors are often simultaneously up- or down-regulated. In vertebrates, sex hormones, such as estrogens and testosterones, orchestrate sexually dimorphic reproductive behaviors by organizing underlying neural circuits during development, and activating these behaviors during adult stages (*McCarthy, 2008*; *McEwen, 1981*; *Tinbergen, 1951*). However, expression of either courtship or aggression to the wrong target sex can be costly. Thus, an animal must be equipped with a neural mechanism that co-regulates these two behaviors while also executing each action selectively toward the right target. How the nervous system manages these two seemingly conflicting demands remains a central question in the neurobiology of social behaviors (*Anderson, 2016*; *Chen and Hong, 2018*; *Li and Dulac, 2018*).

One possible mechanism that can account for the target sex-selective execution of an appropriate behavior is that sex-specific sensory cues are channeled into a dedicated circuit, which in turn triggers a specific behavior toward each sex (*Kohatsu et al., 2011*; *Ruta et al., 2010*; *von Philipsborn et al., 2011*). This hypothetical chain of neurons are sometimes referred to as a 'labeled line' (*Ishii et al., 2017*). In this relatively rigid circuit, behaviorally relevant sensory information represented in the brain is transformed into the motor execution of sexually dimorphic behavior

in a stereotypical manner (*Chen and Hong, 2018*; *Manoli et al., 2013*). Simply put, the labeled line hypothesis predicts that the activation of any neuron that is downstream of sensory perception ('releaser' in a classical term *Tinbergen, 1951*) should in principle trigger the behavior of interest, regardless of the presence of external stimuli (*von Philipsborn et al., 2011*). Sex-specific sensory cues are differentially represented in the brains of both males and females (*Bayless et al., 2019*; *Bergan et al., 2014*; *Datta et al., 2008*; *Haga et al., 2010*; *Kohatsu et al., 2011*; *Kohl et al., 2013*; *Li et al., 2017*; *Remedios et al., 2017*), which is consistent with a simple two-step mechanism that generates sexually dimorphic behaviors: recognition of sex, followed by execution of sex-specific behaviors.

Interestingly, certain populations of neurons are known to be capable of promoting both courtship and aggressive behaviors. In mice, stimulation of neurons in ventromedial hypothalamus (*Lee et al., 2014*) and medial amygdala (*Hong et al., 2014*) induces both behaviors. In fruit flies, neurons located at the posterior medial part of the male brain (commonly referred to as 'P1' or 'pC1' neurons) show a similar dual functionality (*Hoopfer et al., 2015*; *Koganezawa et al., 2016*). Although several mechanisms have been proposed to account for the generation of two behaviors by a seemingly single population of neurons (*Anderson, 2016*; *Koganezawa et al., 2016*), an underlying implicit assumption remains that the activation pattern of the given neurons determines the behavioral outcome. However, for mouse medial amygdala and ventromedial hypothalamus (*Li et al., 2017*; *Remedios et al., 2017*), target sex-specific activation patterns are not hard-wired, but instead emerge as a result of intra-species interactions. Moreover, certain aspects of aggressive behavior induced by the artificial stimulation of mouse ventromedial hypothalamus can be modulated by social contexts, including the sex of target animals (*Lin et al., 2011*; *Yang et al., 2017*). These observations raise the possibility that the hierarchy between the sex-recognition mechanism and the execution mechanism may not be as unidirectional as a 'labeled line' model suggests. Increasing cellular precision for neural manipulation, especially paired with the temporal precision of optogenetics, provides a unique opportunity to address how the sex-recognition mechanism and the execution mechanism interact with one another, and where in the neural circuit behavior specificity toward a given sex is generated.

Here, we addressed the genetic and neural origins of the two mechanisms – sex recognition and execution of behaviors – underlying sexually dimorphic social behaviors of the fruit fly *Drosophila melanogaster*. Instead of sex hormones, *Drosophila* uses two sex-determining genes, *doublesex* (*dsx*) and *fruitless* (*fru*), to specify sexual dimorphisms at the cellular level. We found that optogenetic stimulation of a specific subset of P1/pC1 neurons induces more courtship behavior toward females than toward males. Neuroanatomical sexual dimorphism of this P1/pC1 subset is specified predominantly by *dsx,* and its capacity to promote courtship is retained regardless of *fru* function. However, *fru* is necessary to enhance courtship behavior specifically toward female targets after optogenetic stimulation. We also found evidence that P1/pC1 neurons consist of genetically and functionally heterogeneous populations, one of which requires the contribution of both *dsx* and *fru* for specification of neuroanatomical sexual dimorphism. Lastly, in contrast to courtship behavior, aggressive behavior requires a *fru*-dependent execution mechanism. Our studies illuminate the previously under-appreciated importance of the sex of a target animal as a behaviorally relevant biological variable, and suggest that the relationship between the sex-recognition mechanism and the behavior execution mechanism is interactive rather than hierarchical. For courtship behavior, the function of *dsx* resembles the organizational role of the vertebrate steroid hormone, whereas the function of *fru* can be framed as the activation role. This separation of functions does not extend to aggressive behavior, suggesting that execution mechanisms for different types of sexually dimorphic social behaviors may be specified through separable genetic mechanisms. The neurogenetic approach we employed presents a path to dissect the genetic and circuitry origins of sexually dimorphic social behaviors.

## Results

### The target fly's sex affects the function of social behavior-promoting neurons

Both *dsx* and *fru* genes control the sexually dimorphic specification of *Drosophila* neurons that are critical for sexual behaviors both in males and females (*Dickson, 2008*; *Ellendersen and von Philipsborn, 2017*; *Yamamoto and Koganezawa, 2013*). Namely, a cluster of up to 60 sexually dimorphic neurons located at the posterior medial part of the male *Drosophila* brain, collectively referred to as 'P1' (*Cachero et al., 2010*; *Kimura et al., 2008*; *Kohatsu et al., 2011*; *Lee et al., 2000*; *Pan et al., 2012*; *Ren et al., 2016*; *von Philipsborn et al., 2011*; *Yu et al., 2010*; *Zhou et al., 2015*) or 'pC1' (*Deutsch et al., 2020*; *Kohatsu and Yamamoto, 2015*; *Lee et al., 2002*; *Palavicino-Maggio et al., 2019*; *Ren et al., 2016*; *Rideout et al., 2010*; *Robinett et al., 2010*; *Sanders and Arbeitman, 2008*; *Wang et al., 2020*; *Zhou et al., 2014*) neurons, are considered central for various aspects of male and female reproductive behaviors (*Auer and Benton, 2016*; *Ellendersen and von Philipsborn, 2017*). Artificial activation of male P1/pC1 neurons can induce courtship behavior in the absence of a target fly (*Bath et al., 2014*; *Inagaki et al., 2014*; *Kohatsu et al., 2011*; *von Philipsborn et al., 2011*), suggesting that these neurons can serve as an execution mechanism for courtship. However, activation of certain P1/pC1 subsets are reported to promote aggressive as well as courtship behavior when a male target fly is present (*Hoopfer et al., 2015*; *Koganezawa et al., 2016*), raising a possibility that the function of P1/pC1 neurons is not entirely independent of the target sex.

To address this, we generated 'tester' flies in which the red-shifted channelrhodopsin CsChrimson (*Klapoetke et al., 2014*) was expressed in *dsx*, *fru*-co-expressing neurons by combining *dsx*<sup>GAL4</sup> (*Rideout et al., 2010*) and *fru*<sup>FLP</sup> (*Yu et al., 2010*), which are knock-in alleles of *dsx* and *fru*, respectively. We then used a transgenic element in which the coding sequence for CsChrimson is placed downstream of both the upstream activation sequence (UAS), to which the yeast transcription activator GAL4 binds and drives transcription, and a transcriptional termination signal that can be excised only in the presence of the DNA recombinase Flippase (FLP). Under this configuration, CsChrimson proteins are expressed only in cells in which both GAL4 and FLP are present, which in this case should be the neurons that express *dsx* and *fru* ($dsx^{GAL4} \cap fru^{FLP}$ neurons). We visualized neuronal morphology and soma by detecting immunoreactivity to a red fluorescent protein tdTomato that tags CsChrimson. This approach eliminates possible discrepancies of labeling patterns between marker genes and untagged effector proteins, which cannot be directly visualized.

We observed CsChrimson expression in specific neuronal clusters that correspond to previously characterized *dsx*, *fru*-co-expressing neurons in the male brain and ventral nerve cord, including P1/pC1 neurons (*Figure 1A,B*, *Figure 1—figure supplement 1A*; *Rideout et al., 2007*; *Rideout et al., 2010*; *Sanders and Arbeitman, 2008*; *Zhou et al., 2014*). In contrast, we found very few labeled cells in the female brain (*Figure 1B*, *Figure 1—figure supplement 1B*). We note that the number of cells labeled by our approach was significantly fewer than neurons reported to co-express *dsx* and *fru* by immunohistochemistry (*Rideout et al., 2007*) likely because of a mismatch between knock-in alleles and endogenous gene expression patterns (*Stockinger et al., 2005*; *Yu et al., 2010*), a difference in the expression levels of UAS transgenic elements (*Pfeiffer et al., 2010*; *Pfeiffer et al., 2012*), or an incomplete excision by FLP of the transcriptional termination signals (*Nern et al., 2011*).

We then observed behaviors of these CsChrimson-expressing flies ('tester' flies) toward either a wild-type male or female fly ('target' fly) in a behavioral arena equipped with programmable light-emitting diodes (LEDs) for photostimulation (*Inagaki et al., 2014*; *Figure 1C*). After 1 min of a pre-stimulation period, we applied three blocks of 1 min LED illumination, separated by 2 min inter-stimulus intervals (ISIs) during which the LEDs were turned off (*Figure 1D*). We used three parameters to quantify social interactions: 1) the duration of 'time orienting' toward a target fly (see Materials and methods for the definition), which is a behavior-neutral parameter associated with levels of interaction; 2) number of lunges, which is a highly expressed male-type aggressive action (*Chen et al., 2002*; *Jacobs, 1960*; and 3) duration of unilateral wing extensions (henceforth referred to as wing extensions), which is a frequent action during male-type courtship behavior (*Hall, 1994*; *Murthy, 2010*). For lunges and wing extensions, we developed automated behavioral classifiers using

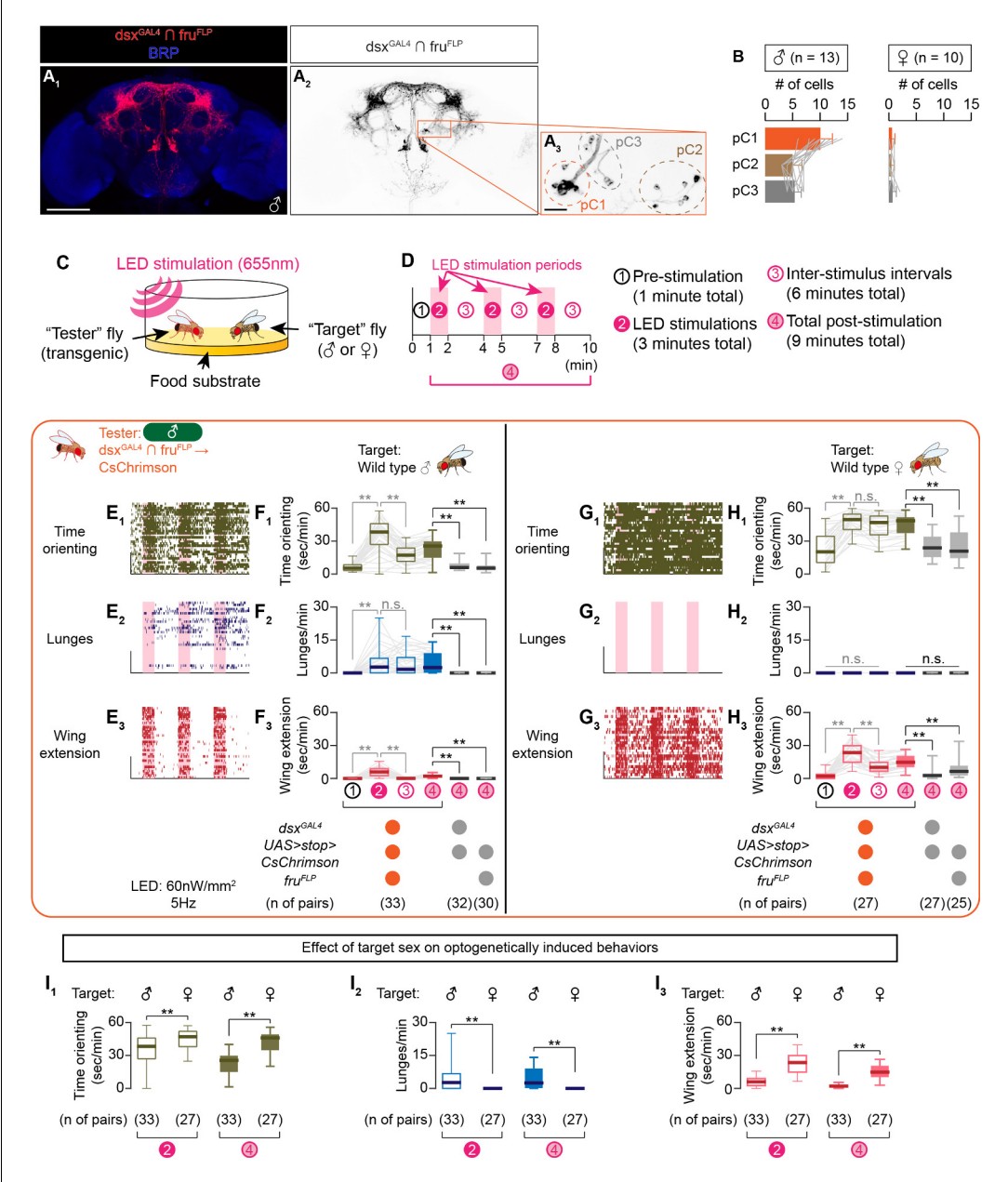

**Figure 1.** Sex of the target fly influences behaviors triggered by the optogenetic activation of social behavior-promoting neurons. (A) Expression of CsChrimson:tdTomato under the control of $dsx^{GAL4}$ and $fru^{FLP}$ (red in $A_1$, black in $A_{2,3}$) in a male brain is visualized together with a neuropil marker BRP (blue in $A_1$) by immunohistochemistry. Labeled cell body clusters are enlarged in $A_3$. Scale bar: 100 μm ($A_1$), 10 μm ($A_3$). (B) Mean number of cell bodies per hemibrain visualized by anti-DsRed antibody in male (left) and female (right) brains. (C) Schematics of the design of behavioral assays. (D) Schematics of the optogenetic stimulation paradigm. Time windows 1–4 represent periods in which behavioral parameters are pooled and calculated in subsequent panels. (E, G) Rasters of behaviors (indicated in left) performed by male tester flies that express CsChrimson:tdTomato under the control of $dsx^{GAL4}$ and $fru^{FLP}$. Pink bar: LED-on periods, horizontal bar: 10 min (also see D), vertical bar: 10 flies. LED stimulation condition is indicated at the bottom. (F, H): Boxplots of time orienting ($F_1$, $H_1$), lunges ($F_2$, $H_2$), and wing extension ($F_3$, $H_3$) by the tester flies during the time windows 1–4 (see D). Testers' genotypes and pair numbers are indicated below the plots. Gray lines represent single testers. In gray: **p<0.01, n.s. p>0.05 (Kruskal-Wallis one-way ANOVA and post-hoc Wilcoxon signed rank test). In black: **p<0.01, n.s., p>0.05 (Kruskal-Wallis one-way ANOVA and post-hoc Mann-Whitney U-test). Target flies are either wild-type group-housed males (E, F) or mated females (G, H). (I) Comparison of time orienting ($I_1$), lunges ($I_2$), and wing extension ($I_3$) performed by tester flies that express CsChrimson:tdTomato under the control of $dsx^{GAL4}$ and $fru^{FLP}$ in males (data replotted from F, H) toward male or female target flies (indicated above). Number of pairs tested and time windows compared are indicated below the panels. **p<0.01 (Mann-Whitney U-test). For detailed methods to quantify behaviors, see Materials and methods.

The online version of this article includes the following figure supplement(s) for figure 1:

*Figure 1 continued on next page*

the machine learning-based platform JAABA (*Kabra et al., 2013*; *Figure 1—figure supplement 2A*, B$_{1-2}$; see also Materials and methods for details).

When the target fly was male, optogenetic activation of dsx$^{GAL4}$ ∩ fru$^{FLP}$ neurons induced both wing extensions and lunges across a range of stimulation LED frequencies (*Figure 1E,F*, *Figure 1—figure supplement 3A*; see also *Video 1* – part 1). Toward a female target fly, the same optogenetic activation induced robust wing extensions (*Figure 1G3–H3*, *Figure 1—figure supplement 3B3*) but no lunges (*Figure 1G$_2$, H$_2$, I$_2$*, *Figure 1—figure supplement 3B2*) (see also *Video 1* – part 2). The presence of female target flies induced more wing extensions from tester flies than did the presence of male target flies (*Figure 1I$_3$*). Optogenetic activation of dsx$^{GAL4}$ ∩ fru$^{FLP}$ neurons also increased time orienting toward females more than it did toward males (*Figure 1E$_1$, I$_1$*), suggesting that the lack of lunges toward female targets is not due to the lack of opportunities to interact (due to immobility, for instance). We therefore conclude that activation of dsx$^{GAL4}$ ∩ fru$^{FLP}$ neurons can induce more wing extensions toward female than toward male targets, and lunges only if a male target fly is present. In other words, the activation of this neuronal population by itself does not determine the behavior of the tester fly. Even under optogenetic manipulations, behaviors of tester flies are influenced by target flies, likely through sex-specific sensory cues or subtle behaviors of target flies that are currently not quantified.

NP2631 ∩ dsx$^{FLP}$ neurons contain subpopulations of the P1/pC1 cluster that promote both courtship and aggression dsx$^{GAL4}$ ∩ fru$^{FLP}$ neurons in the brain contain several distinct clusters (*Figure 1A$_3$*; *Rideout et al., 2007*; *Zhou et al., 2014*). As discussed above, the P1/pC1 cluster is the most likely neural substrate among them that trigger the above-described social behaviors (*Auer and Benton, 2016*; *Ellendersen and von Philipsborn, 2017*; *Koganezawa et al., 2016*; *Pan and Baker, 2014*; *Pan et al., 2011*) (see also *Deutsch et al., 2019*; *Kohatsu and Yamamoto, 2015* for contributions of the pC2 cluster). Our finding that activation of dsx$^{GAL4}$ ∩ fru$^{FLP}$ neurons as a whole triggers social behaviors in a target sex-dependent manner prompted us to ask whether the previously reported 'dual functionality' of P1/pC1 subsets (e.g., neurons that induce both courtship and aggression upon stimulation; *Hoopfer et al., 2015*; *Koganezawa et al., 2016*) is also influenced by the sex of target flies.

We first focused on the neurons that are labeled by the combination of an enhancer trap GAL4 line NP2631 and dsx$^{FLP}$, a knock-in allele of *dsx* (*Rezával et al., 2014*) (we hereafter refer to this population as 'NP2631 ∩ dsx$^{FLP}$ neurons'). This population includes a specific subset of P1/pC1 neurons that is reported to promote both courtship and aggression toward a male target (*Koganezawa et al., 2016*). Consistent with this, we found that the optogenetic activation of NP2631 ∩ dsx$^{FLP}$ neurons robustly increased both wing extensions and lunges toward male target flies (*Figure 2A*, *Figure 2—figure supplement 1A*; see also *Video 2* – part 1). Interestingly, however, the same manipulation did not trigger lunges when the target flies were females (*Figure 2B$_2$*, *Figure 2—figure supplement 1B*). Instead, activation of NP2631 ∩

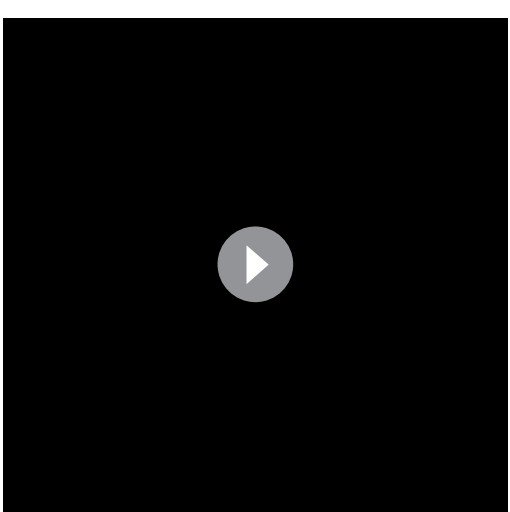

**Video 1.** Representative behavior of a male tester fly that expresses. CsChrimson:tdTomato under the control of *dsx$^{GAL4}$* and *fru$^{FLP}$* toward a wild-type male (Part 1) or a wild-type female (Part 2) target fly, at the onset and offset of LED stimulation.
https://elifesciences.org/articles/52701#video1

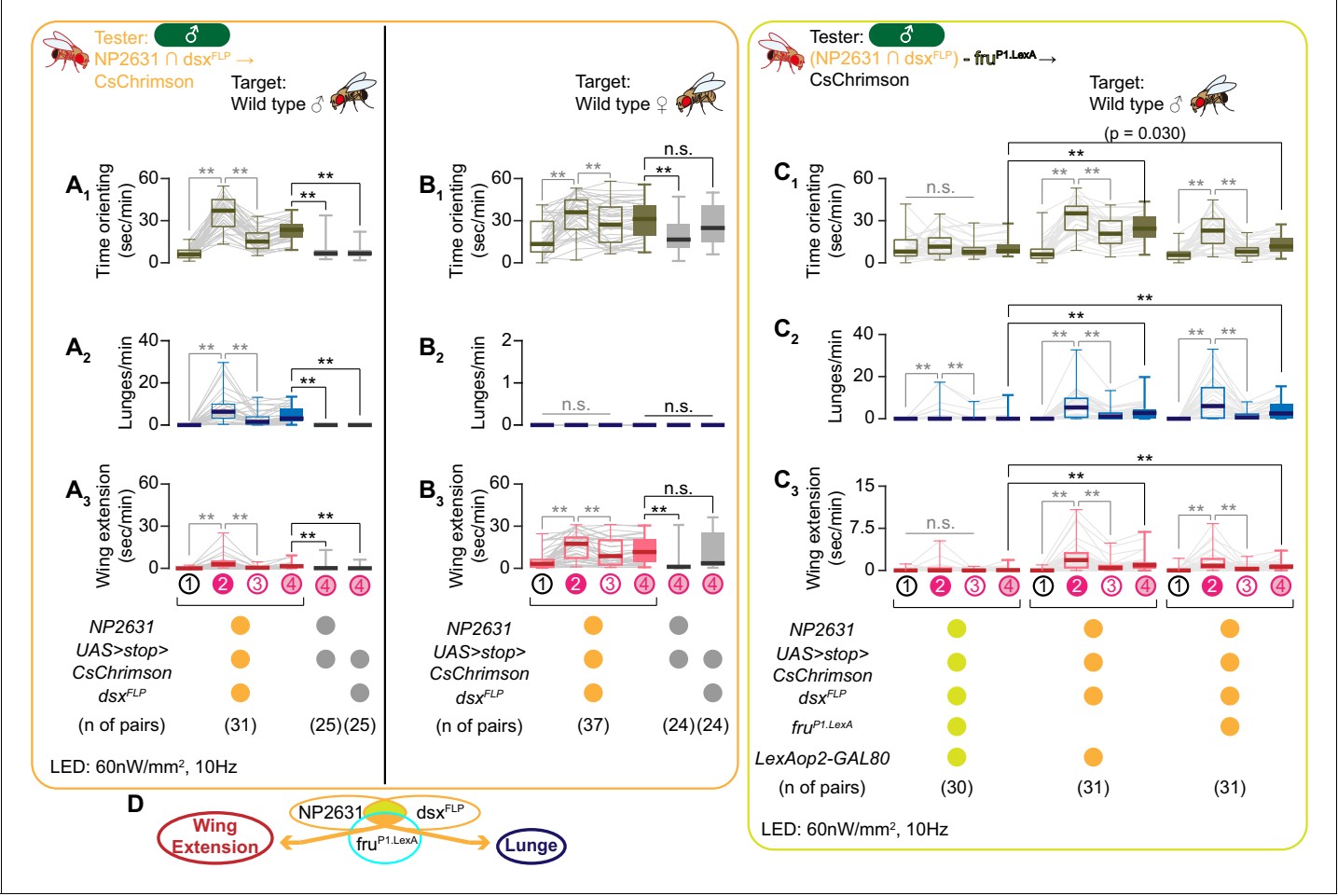

**Figure 2.** NP2631 ∩ dsx$^{FLP}$ neurons that express both *dsx* and *fru* can promote courtship and aggressive behaviors. (A–C): Boxplots of time orienting (A$_1$–C$_1$), lunges (A$_2$–C$_2$), and wing extension (A$_3$–C$_3$) by the tester flies during the time windows 1–4 (see *Figure 1D*). Testers' genotypes and pair numbers are indicated below the plots. Gray lines represent single testers. Target flies are either group-housed wild-type males (A, C) or mated females (B). Gray lines represent single testers. In gray: **p<0.01, n.s. p>0.05 (Kruskal-Wallis one-way ANOVA and post-hoc Wilcoxon signed rank test). In black: **p<0.01, n.s. p>0.05 (Kruskal-Wallis one-way ANOVA and post-hoc Mann-Whitney U-test). (D). A schematic summary. A subset of NP2631 ∩ dsx$^{FLP}$ neurons that express *fru* (labeled by the *fru$^{P1.LexA}$* allele) (orange) can promote both lunges and wing extensions.

The online version of this article includes the following figure supplement(s) for figure 2:

**Figure supplement 1.** Detailed description of behaviors induced by the optogenetic activation of NP2631 ∩ dsx$^{FLP}$ neurons.

**Figure supplement 2.** A detailed characterization of NP2631 ∩ dsx$^{FLP}$ neurons.

**Figure supplement 3.** Behaviors of target flies remain largely unaffected by the optogenetic stimulation of NP2631 ∩ dsx$^{FLP}$ neurons in tester flies.

dsx$^{FLP}$ neurons in the presence of female target flies induced more wing extensions during LED stimulations than during ISIs (*Figure 2B$_3$*; see also *Video 2* – Part 2). Time orienting toward both male and female targets increased after optogenetic stimulation of NP2631 ∩ dsx$^{FLP}$ neurons (*Figure 2A$_1$, B$_1$*), excluding the possibility that tester males did not have an opportunity to lunge toward female targets. These behavioral changes were largely consistent across different stimulation LED frequencies (*Figure 2—figure supplement 1C*). However, in contrast to the previous observation (*Koganezawa et al., 2016*), we did not observe a lunge-like behavior or change in speed when we optogenetically stimulate NP2631 ∩ dsx$^{FLP}$ neurons in the absence of target flies (*Figure 2—figure supplement 1D*). Thus, the function of NP2631 ∩ dsx$^{FLP}$ neurons depends on the sex of target flies.

With the genetic reagents we used (*UAS > stop > CsChrimson:tdTomato* as a reporter as well as an effector element: see *Supplementary file 1* for details), we consistently detected expression of CsChrimson:tdTomato in a single pair of neuronal clusters located at the medial posterior region of

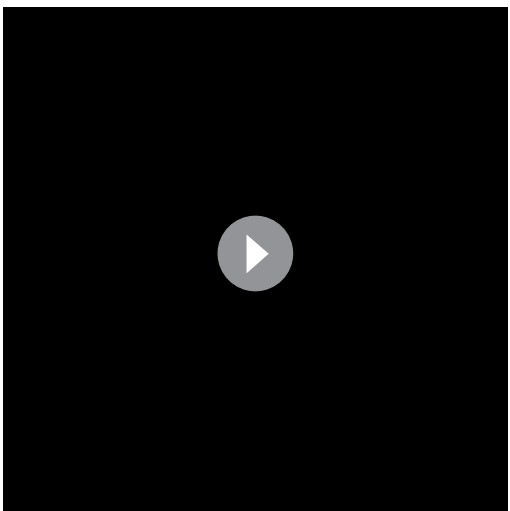

**Video 2.** Representative behavior of a male tester fly that expresses. CsChrimson:tdTomato under the control of *NP2631* and *dsx^FLP^* toward a wild-type male (Part 1) or a wild-type female (Part 2) target fly, at the onset and offset of LED stimulation.
https://elifesciences.org/articles/52701#video2

male brains (23 out of 23 brains examined in *Figure 2—figure supplement 2D1*; orange circles in *Figure 2—figure supplement 2A1*; see also *Video 3*). As was reported in *Koganezawa et al. (2016)*, we found that nearly all of these cell bodies express Dsx proteins, and a subset of them also co-express FruM proteins (*Figure 2—figure supplement 2B,C*), indicating that they belong to the 'P1/pC1' neuronal cluster. In addition, we occasionally observed a neuron with a large cell body and a tract that is not shared with the P1/pC1 cluster. This neuron has a prominent arborization in the contralateral ventromedial neuropil and a long descending fiber through the entire ventral nerve cord (*Figure 2—figure supplement 2A2*). We could not resolve its arborizations that overlap with those of the above-mentioned P1/pC1 cluster. The overall morphology resembles the previously documented *dsx*-expressing pMN1 neuron (*Deutsch et al., 2019*; *Bogovic et al., 2018*; *Robinett et al., 2010*), but unlike the descending neurons we found, the arborization in the ventromedial neuropil seems absent in pMN1 (*Kimura et al., 2015*). This descending neuron was present in up to 35% of male brains, all of which had only one such neuron in one of the two hemispheres (*Figure 2* – figure supplement $D_1$). Importantly, we observed an increase in both lunges and wing extensions upon optogenetic stimulation of NP2631 ∩ dsx^FLP^ neurons far more consistently than the presence of the descending neuron (*Figure 2—figure supplements 2D2* and *3*). Based on this observation, we conclude that the behavioral effects by the activation of NP2631 ∩ dsx^FLP^ neurons are due to its P1/pC1 neuronal cluster. For the sake of simplicity, we will hereafter use 'NP2631 ∩ dsx^FLP^ neurons' to refer to its P1/pC1 cluster. We do not exclude the possibility that the occasionally labeled descending neuron can contribute to social behaviors as well.

The previous report (*Koganezawa et al., 2016*) concluded that the *fru*-expressing subset of NP2631 ∩ dsx^FLP^ neurons specifically promotes courtship, whereas the *fru*-negative subset specifically promotes aggression. This division of functions is in contrast to our observation with dsx^GAL4^ ∩ fru^FLP^ neurons. We therefore took the same genetic approach as the previous report (*Koganezawa et al., 2016*) to limit the expression of CsChrimson in the neurons that do not express *fru^P1.LexA^* (a knock-in allele that expresses a bacterial transcription factor LexA in place of male-specific *fru* isoforms) (*Mellert et al., 2010*). We confirmed that this genetic approach reduced the number of labeled NP2631 ∩ dsx^FLP^ neurons (3.1 ± 1.3, n = 16) to a degree that is predicted from the result of FruM immunohistochemistry within this cluster (*Figure 2—figure supplement 2E–H*). When the expression of CsChrimson was suppressed in *fru^P1.LexA^*-expressing subpopulations, the amount of optogenetically induced lunges and wing extensions were both significantly decreased compared to controls that have CsChrimson in all NP2631 ∩ dsx^FLP^ neurons (*Figure 2C*). The amount of lunges during LED stimulations in this genotype was still significantly higher than those before LED stimulations or during ISIs (*Figure 2C2*), which can be due to some *fru^P1.LexA^*-negative neurons that specifically

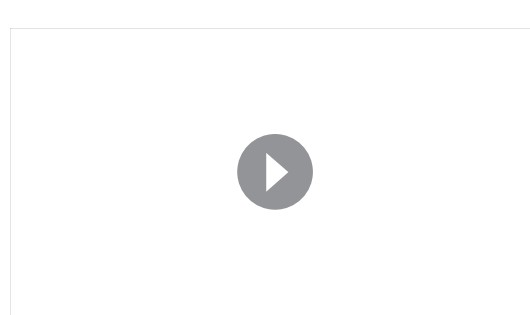

**Video 3.** 3D-rendered average image of NP2631 ∩ dsx^FLP^ neurons in male (green) and in female (magenta). Gray represents a standard unisex *Drosophila* brain (*Bogovic et al., 2018*).
https://elifesciences.org/articles/52701#video3

promote aggression (*Koganezawa et al., 2016*), or incomplete suppression of NP2631 in $fru^{P1.LexA}$-expressing neurons in some animals. Regardless, the significant reduction of lunges and wing extensions in the absence of the $fru^{P1.LexA}$-expressing subgroup favors the idea that the *fru*-expressing subpopulation (NP2631 $\bigcap$ dsx$^{FLP}$ $\bigcap$ $fru^{P1.LexA}$) is capable of promoting both behaviors. These results led us to conclude that NP2631 $\bigcap$ dsx$^{FLP}$ neurons contain *dsx* and *fru*-expressing subset of P1/pC1 neurons that can promote courtship and aggressive behaviors depending on the sex of target flies (*Figure 2D*).

Courtship or aggressive behaviors of target flies can feed back to tester flies and change their behaviors. Therefore, we also quantified behaviors of male and female target flies while the NP2631 $\bigcap$ dsx$^{FLP}$ neurons of tester flies were optogenetically activated. For this purpose, we created a classifier for headbutt, a female-type aggressive action (*Nilsen et al., 2004*; *Ueda and Kidokoro, 2002*; *Figure 1—figure supplement 1A*, B$_3$). We observed that the behavioral levels of group-housed target flies, for the most part, remained low regardless of the genotypes of tester flies (*Figure 2—figure supplement 3A,B*). Although male target flies under this condition showed a statistically significant increase in lunges (*Figure 2—figure supplement 3A2*), the magnitude was very small. We therefore conclude that target flies do not alter tester flies' behaviors by actively performing courtship or aggressive behaviors. For clarity, we henceforth only show behaviors of the tester flies.

## *Dsx* specifies NP2631 $\bigcap$ dsx$^{FLP}$ neurons

Considering the importance of *dsx* and *fru*-co-expressing neurons on both courtship and aggressive behaviors, we next asked the role of the two sex-determining genes on the specification of NP2631 $\bigcap$ dsx$^{FLP}$ neurons. As a result of sex-specific splicing (*Figure 3A*), males (whose sex chromosome composition is XY) transcribe male-specific mRNA isoforms of *dsx* (dsxM) and *fru* (fruM) (green in *Figure 3B*), whereas females (whose sex chromosome composition is XX) have female-specific mRNA isoforms of *dsx* (dsxF) and *fru* (fruF) (magenta in *Figure 3B*). To dissociate the role of *dsx* and *fru*, we took advantage of two genetic alleles of *fru*, $fru^M$ and $fru^F$ (*Demir and Dickson, 2005*). These *fru* alleles force male and female-specific splicing, respectively, regardless of the sex chromosome composition. As a result, a chromosomal male (XY) that carries the $fru^F$ allele expresses dsxM and fruF (brown in *Figure 3B*), whereas a chromosomal female (XX) that carries the $fru^M$ allele expresses dsxF and fruM (blue in *Figure 3B*), dissociating the sex-specific isoforms of *dsx* and *fru*. Because the *dsx* isoform determines almost all somatic gender characteristics (*Baker and Ridge, 1980*; *Nagoshi and Baker, 1990*), chromosomally male $fru^F$-bearing gynandromorphs ('fruF males') appear anatomically male, whereas chromosomally female $fru^M$-bearing gynandromorphs ('fruM females') appear anatomically female (*Demir and Dickson, 2005*). These four genotypes exhaust all possible combinations of sex-specific *dsx* and *fru* splicing species, affording us an opportunity to address which 'genetic sex' (that is, sex-specific splicing of *dsx* or *fru*) is responsible for observed sexual dimorphism.

Sexual dimorphism of NP2631 $\bigcap$ dsx$^{FLP}$ neurons has not been characterized in detail (*Koganezawa et al., 2016*). We observed that more NP2631 $\bigcap$ dsx$^{FLP}$ neurons are labeled in males than in females (*Figure 3G*). NP2631 $\bigcap$ dsx$^{FLP}$ neurons in both sexes extend a single tract that innervates the so-called lateral protocerebral complex (*Yu et al., 2010*) including commissural projections at the superior dorsal protocerebrum (*Figure 3C$_2$, F$_2$*). To better visualize the population-level sex differences in neuroanatomy, we registered the z-stack images of immunohistochemically labeled brains into a standard unisex *Drosophila* brain template (*Bogovic et al., 2018*). Visual inspection of 3D-reconstructed standardized neurons and quantification of specific neuronal segments revealed several notable sexual dimorphisms. Firstly, male neurons have thicker projections at the lateral junctions (orange in *Figure 3H,K*; *Figure 3L*; *Yu et al., 2010*). Secondly, male neurons consistently have a lateral segment branching within the 'ring' projection (*Yu et al., 2010*) ('lateral vertical projection') that female neurons almost completely lack (green in *Figure 3H,K*; *Figure 3M*). Thirdly, female neurons have more extensive projections at the superior medial part of the brain ('superior medial projection') (magenta in *Figure 3H,K*; *Figure 3N*) (see also *Video 3*). Thus, NP2631 $\bigcap$ dsx$^{FLP}$ neurons are sexually dimorphic.

We next asked whether *dsx* or *fru* is responsible for this sexual dimorphism. In fruF males, NP2631 $\bigcap$ dsx$^{FLP}$ neurons (*Figure 3E*) are similar to male neurons in cell body number and all the three sexually dimorphic neuroanatomical characteristics listed above (*Figure 3G,J,L–N* (brown); see

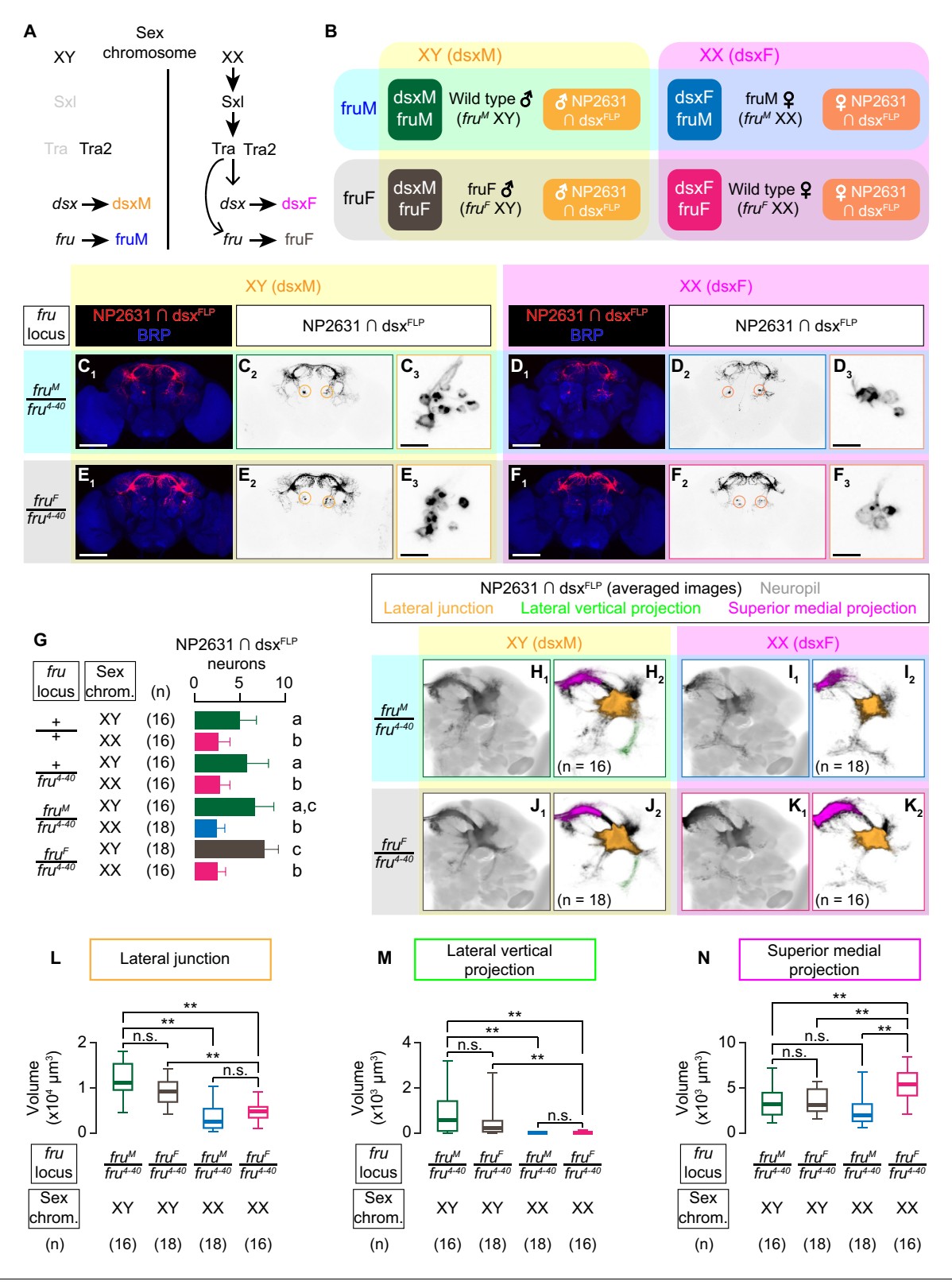

**Figure 3.** *dsx* specifies the sexual dimorphism of NP2631 ∩ dsx^FLP neurons. (**A**) Schematics of the sex-determination pathway in *Drosophila*. (**B**) Schematic of the four sex genotypes defined by *dsx* and *fru* splicing, and how NP2631 ∩ dsx^FLP neurons are specified in each genotype (see following panels for details). (**C–F**) Expression of CsChrimson:tdTomato under the control of *NP2631* and *dsx^FLP* (red in C₁–F₁, black in C₂,₃-F₂,₃) in brains of a male (**C**), fruM female (**D**), fruF male (**E**), and female (**F**) is visualized together with a neuropil marker BRP (blue) by immunohistochemistry. Circle: soma

*Figure 3 continued on next page*

Figure 3 continued

(right cluster is enlarged in $C_3$–$F_3$). Scale bar: 100 µm ($C_1$–$F_1$), 10 µm ($C_3$–$F_3$). (G) Mean number of cell bodies per hemibrain visualized by anti-DsRed antibody in each genotype represented in C–F) and *Figure 3—figure supplement 1A–D*. (H–K) Z-projection of the registered and averaged images of CsChrimson:tdTomato expression under the control of *NP2631* and *dsx*$^{FLP}$ (black in $H_1$–$K_1$) in male (H), fruM female (I), fruF male (J), and female (K). A part of the standard *Drosophila* brain is shown in gray in $H_2$–$K_2$. Number of used hemibrains are indicated in $H_2$–$K_2$). Lateral junction (orange), lateral vertical projection (green), and superior medial projection (magenta) are segmented and overlaid in $H_2$-$K_2$. L-M: Boxplot of volumes of lateral junction (L), lateral vertical projection (M), and superior medial projections (N). Genotypes and number of hemibrains are indicated below the plot. **$p<0.01$, n.s. $p>0.05$ (Kruskal-Wallis one-way ANOVA and post-hoc Wilcoxon signed rank test).

The online version of this article includes the following figure supplement(s) for figure 3:

**Figure supplement 1.** Heterozygosity of the *fru*$^{4-40}$ allele leaves the neuroanatomy and function of NP2631 ∩ dsx$^{FLP}$ neurons largely unaltered.

---

also *Video 4*), while NP2631 ∩ dsx$^{FLP}$ neurons in fruM females (*Figure 3D*) are similar to those of females in cell body number and two of the three characteristics (*Figure 3G,I,L–M* (blue); see also *Video 5*). The only exception was the size of superior medial projection, which in this genotype was comparable to that in males (*Figure 3N*). These observations suggest that morphological sexual dimorphisms of NP2631 ∩ dsx$^{FLP}$ neurons are established predominantly by the sexual dimorphism of *dsx*. Complete transformation of this class of neurons to the female type may also require the cooperative contribution of *fru*. In addition, we cannot exclude the possibility that subset-specific neuroanatomical changes in either fruF males or fruM females escaped our detection.

We used the *fru*$^M$ and *fru*$^F$ alleles in trans with a male isoform-specific deletion allele, *fru*$^{4-40}$, following previous studies (*Rideout et al., 2007*; *Vrontou et al., 2006*). This configuration was necessary to eliminate potential confounds by incomplete dominance of the *fru*$^M$ allele, which cannot be made homozygous because females that carry the *fru*$^M$ allele do not mate (see also *Wohl et al., 2020* for discussion regarding incomplete dominance of the *fru*$^M$ allele). Presence of the *fru*$^{4-40}$ allele by itself did not affect overall sexually dimorphic characteristics of NP2631 ∩ dsx$^{FLP}$ neurons (*Figure 3G*, *Figure 3—figure supplement 1A–H*). Activation of this population induced qualitatively similar behavioral changes in males with the *fru* locus of +/+, +/*fru*$^{4-40}$, or *fru*$^M$/*fru*$^{4-40}$ (*Figure 3—figure supplement 1I*).

## Activation of NP2631 ∩ dsx$^{FLP}$ neurons in the absence of fruM induces courtship

The presence of male-like NP2631 ∩ dsx$^{FLP}$ neurons in fruF males raises a question about the behavioral role of these neurons in this genotype. fruF males are defective in enhancing courtship specifically toward conspecific females, but they are still capable of executing courtship actions (*Demir and Dickson, 2005*; *Fan et al., 2013*; *Pan and Baker, 2014*; *Shirangi et al., 2006*; *Villella et al., 1997*). These observations suggest that the courtship deficits in fruF males are not necessarily due to the absence of the courtship execution mechanism, but specifically in the mechanism to recognize proper targets for courtship. Importantly, activation of the entire *dsx*-expressing neurons in fruF males increases courtship behavior (*Pan et al., 2011*), suggesting that neural substrates for the courtship execution mechanism are present in fruF males. We asked whether NP2631 ∩ dsx$^{FLP}$ neurons in fruF males might be a part of this mechanism.

Optogenetic activation of NP2631 ∩ dsx$^{FLP}$ neurons in fruF males increased both time orienting (*Figure 4A$_1$, B$_1$*) and wing extensions (*Figure 4A$_3$, B$_3$*), toward both males and females (*Figure 4—figure supplement 1A,B*; see also *Video 6*). This result suggests that this neuronal population is indeed capable of generating at least an aspect of male-type courtship behavior in a fruM-independent manner. Interestingly, in fruF males, the amount of wing extensions

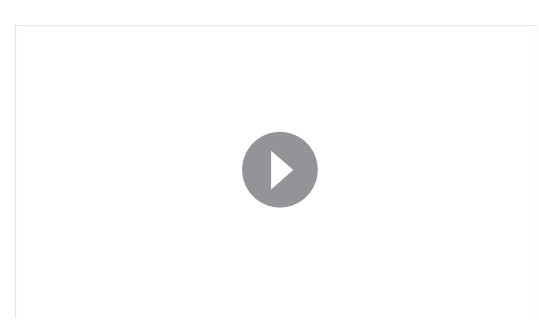

**Video 4.** 3D-rendered average image of NP2631 ∩ dsx$^{FLP}$ neurons in male (green; duplication from *Video 3*) and in fruF male (yellow). Gray represents a standard unisex *Drosophila* brain (*Bogovic et al., 2018*).

https://elifesciences.org/articles/52701#video4

toward male and female target flies was comparable after optogenetic activation of NP2631 ∩ dsx$^{FLP}$ neurons (*Figure 4C*, right). This is in contrast to the situation in regular males, which performed more wing extensions toward females (*Figure 4C*, left). Nonetheless, optogenetic activation of NP2631 ∩ dsx$^{FLP}$ neurons in regular males and in fruF males induced a similar amount of wing extensions toward male targets (*Figure 4C*), suggesting that the lack of fruM specifically abrogates tester flies' capacity to enhance the intensity of courtship behavior selectively toward females. fruF males do not show lunges (*Lee and Hall, 2000*; *Vrontou et al., 2006*), which suggests that the execution mechanism for male-type aggressive behavior depends on fruM. In fact, the same optogenetic manipulation induced almost no lunges toward either target sex (*Figure 4A₂, B₂*; see also *Video 6*). We therefore conclude that sexual dimorphism in *dsx* and *fru* genes have different impacts on the function of NP2631 ∩ dsx$^{FLP}$ neurons. For courtship, *dsx* specifies NP2631 ∩ dsx$^{FLP}$ neurons as an execution component, while amplification of this function specifically toward female targets is fruM-dependent (*Figure 4D*). On the other hand, a fruM-dependent mechanism is necessary for NP2631 ∩ dsx$^{FLP}$ neurons to promote male-type aggressive behaviors, even though gross neuroanatomy of this population is specified primarily by *dsx*.

## P1$^a$ neurons are morphologically and functionally distinct from NP2631 ∩ dsx$^{FLP}$ neurons

A P1/pC1 subset that promotes both courtship and aggressive behaviors is also labeled by the P1$^a$ split GAL4 line (*Hoopfer et al., 2015*; *Inagaki et al., 2014*), which is the genetic intersection of two promoter fragments *R15A01* and *R71G01*. Interestingly, we found that LexA lines made from these two promoters in the male brains labeled largely separate populations from NP2631 ∩ dsx$^{FLP}$ neurons (*Figure 5A–D*). Projections of standardized P1$^a$ (*Figure 5E*) and NP2631 ∩ dsx$^{FLP}$ neurons (*Figure 5F*) only partially overlap (*Figure 5G*; see also *Video 7*). Moreover, the temporal dynamics of behaviors induced by optogenetic activation of NP2631 ∩ dsx$^{FLP}$ and P1$^a$ neurons are different. Activation of NP2631 ∩ dsx$^{FLP}$ neurons increased both lunges and wing extensions mostly during LED stimulations, coinciding with an increase in time orienting toward target flies (*Figure 2A,B*, *Figure 2—figure supplement 1C*). In contrast, stimulation of P1$^a$ neurons increased lunges toward males and wing extensions toward females mostly during ISIs (*Figure 5—figure supplement 1A₂, B₃*) (*Hoopfer et al., 2015*; *Jung et al., 2020*). The increase of wing extensions in the presence of male target flies, which is largely confined to LED stimulations, was not accompanied by the increase of time orienting (*Figure 5—figure supplement 1A₁, A₃*). In addition, while we found that all P1$^a$ neurons are immunoreactive to Dsx (*Figure 5—figure supplement 1C,D*), intersectional combination of *dsx$^{FLP}$* and *P1$^a$-GAL4* labeled almost no neurons (*Figure 5—figure supplement 2A*). This can be because of the low efficacy of *dsx$^{FLP}$* in some *dsx*-expressing cells. Regardless, these differences further support the idea that NP2631 ∩ dsx$^{FLP}$ and P1$^a$ neurons contain non-overlapping and functionally distinct populations.

Optogenetic stimulation of the *fru$^{FLP}$*-expressing subpopulation within P1$^a$ neurons (*R15A01* ∩ *R71G01* ∩ *fru$^{FLP}$* 'triple intersection'; *Figure 5—figure supplement 2A*) promoted both lunges and wing extensions toward male target flies (*Figure 5—figure supplement 2B*), similar to the entire P1$^a$ population (*Figure 5—figure supplement 1A*). Previously, the artificial activation of *R15A01* ∩ *fru$^{FLP}$* and *R71G01* ∩ *fru$^{FLP}$* neurons was reported to promote only lunges (*Hoopfer et al., 2015*; *Watanabe et al., 2017*). However, P1$^a$ neurons are also known to alter the relative ratio of courtship and aggressive behaviors depending on the activation level (*Hoopfer et al., 2015*), leaving a possibility that a difference in activation strength due to different stimulation conditions or genetic reagents used may account for this perceived discrepancy. Consistent with the result of the entire P1$^a$ population, the P1$^a$ ∩ fru$^{FLP}$ neurons are morphologically distinct from NP2631 ∩ dsx$^{FLP}$ neurons (*Figure 5—figure supplement 2C–E*).

In addition, we found that *R15A01* and *R71G01* promoters label largely non-overlapping neurons with another P1/pC1 subset NP2631 ∩ fru$^{FLP}$ neurons (*von Philipsborn et al., 2011*; *Yu et al., 2010*; *Figure 5—figure supplement 3A–D*), underscoring the potential diversity among P1/pC1 neurons (*Costa et al., 2016*; *Zhang et al., 2018*; see also Discussions).

Together, our results suggest that P1$^a$ neurons contain a subset of *dsx* and *fru*-co-expressing P1/pC1 neurons that are morphologically and functionally distinct from NP2631 ∩ dsx$^{FLP}$ neurons. While we do not exclude the possibility that behavior-specific subsets exist within each population, we conclude that activation of distinct *dsx* and *fru*-co-expressing P1/pC1 subsets can promote courtship

and aggression with different dynamics. Activation of these neurons does not necessarily serve as a rigid command for specific behaviors, but instead is modulated by another mechanism that enhances courtship or aggression according to the sex of target flies.

## Cooperative roles of *dsx* and *fru* on the specification of P1ᵃ neurons

P1ᵃ neurons appeared only in male flies (*Hoopfer et al., 2015*; *Figure 6A,D*). The lack of labeling in females likely reflects the absence of P1ᵃ neurons, since *dsx* and *fru*-co-expressing neurons in the posterior part of the brain are known to undergo DsxF-dependent apoptosis (*Kimura et al., 2008*; *Sanders and Arbeitman, 2008*). If this is the case, we expect that P1ᵃ neurons are specified in fruF males (which express dsxM), but not in fruM females (which express

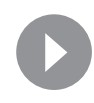

**Video 5.** 3D-rendered average image of NP2631 ∩ dsx^FLP neurons in female (magenta, duplication from *Video 3*) and in fruM female (blue). Gray represents a standard unisex *Drosophila* brain (*Bogovic et al., 2018*).
https://elifesciences.org/articles/52701#video5

dsxF), similar to male-type NP2631 ∩ dsx^FLP neurons.

Indeed, we found P1ᵃ neurons in fruF males (*Figure 6C*), but not in fruM females (*Figure 6B*). The number of P1ᵃ cell bodies was comparable between males and fruF males (*Figure 6E*). Interestingly, registered P1ᵃ neurons in male and fruF male brains showed quantitative differences in their neuroanatomy. Most notably, innervations of P1ᵃ neurons in fruF males were generally thinner. We specifically found that the volume of projections at the lateral junction was smaller in fruF males than in males (*Figure 6F,G,H*). In contrast, neural processes emanating from the posterior side of the lateral junction ('dorsal posterior projection') was more prominent in fruF males than in males (*Figure 6F,G, I*; see also *Video 8*). These observations indicate that sexually dimorphic characteristics of the P1ᵃ neurons are specified by both *dsx* and *fru*. Such cooperative roles of *dsx* and *fru* are reminiscent of the previous report on the roles of the two genes in a *fru*-expressing P1/pC1 neuronal cluster labeled by different genetic reagents (*Kimura et al., 2008*).

We also asked whether P1ᵃ neurons in fruF males can influence male-type courtship or aggressive behaviors. Similar to the case in NP2631 ∩ dsx^FLP neurons, presence of the *fru^4-40* allele did not affect the overall morphology of P1ᵃ neurons (*Figure 6E*, *Figure 6—figure supplement 1A–F*) or the behavioral effect of optogenetic stimulations (*Figure 6—figure supplement 1G*), although lunges toward males were not as robustly induced in the presence of the *fru^4-40* allele. Optogenetic activation of P1ᵃ neurons in fruF males robustly increased time orienting toward both male and female target flies (*Figure 6—figure supplement 2A1*, B₁), but its capacity to promote wing extensions was severely impaired. We observed sporadic wing extensions during LED stimulations and ISIs, which were significantly more frequent than wing extensions performed by genetic controls (*Figure 6—figure supplement 2A3*, B₃). However, the amount of optogenetically induced wing extensions was significantly less than what P1ᵃ neurons in normal males induced toward male targets (*Figure 6—figure supplement 2C*), which is in contrast to the situation in male NP2631 ∩ dsx^FLP neurons.

Similar to NP2631 ∩ dsx^FLP neurons, optogenetic activation of P1ᵃ neurons in fruF males induced virtually no lunges (*Figure 6—figure supplement 2A2*, B₂). Although it is possible that a cell-specific rescue of fruM within P1ᵃ neurons in otherwise fruF males may restore their capacity to promote both courtship and aggression, we have evidence suggesting that the male type aggression-promoting function of both NP2631 ∩ dsx^FLP and P1ᵃ neurons requires other fruM-expressing neurons (see discussion and *Wohl et al., 2020* for details).

Overall, these results demonstrate that NP2631 ∩ dsx^FLP and P1ᵃ neurons are distinct not only in neuroanatomy and behavioral roles, but also in the genetic specification mechanisms. Genetic and functional diversity within the *dsx* and *fru*-co-expressing P1/pC1 cluster suggests that *Drosophila* social behavior can be tuned at a multitude of circuit nodes. Our result also underscores the importance of the precise identification of cell types at the focus of a study.

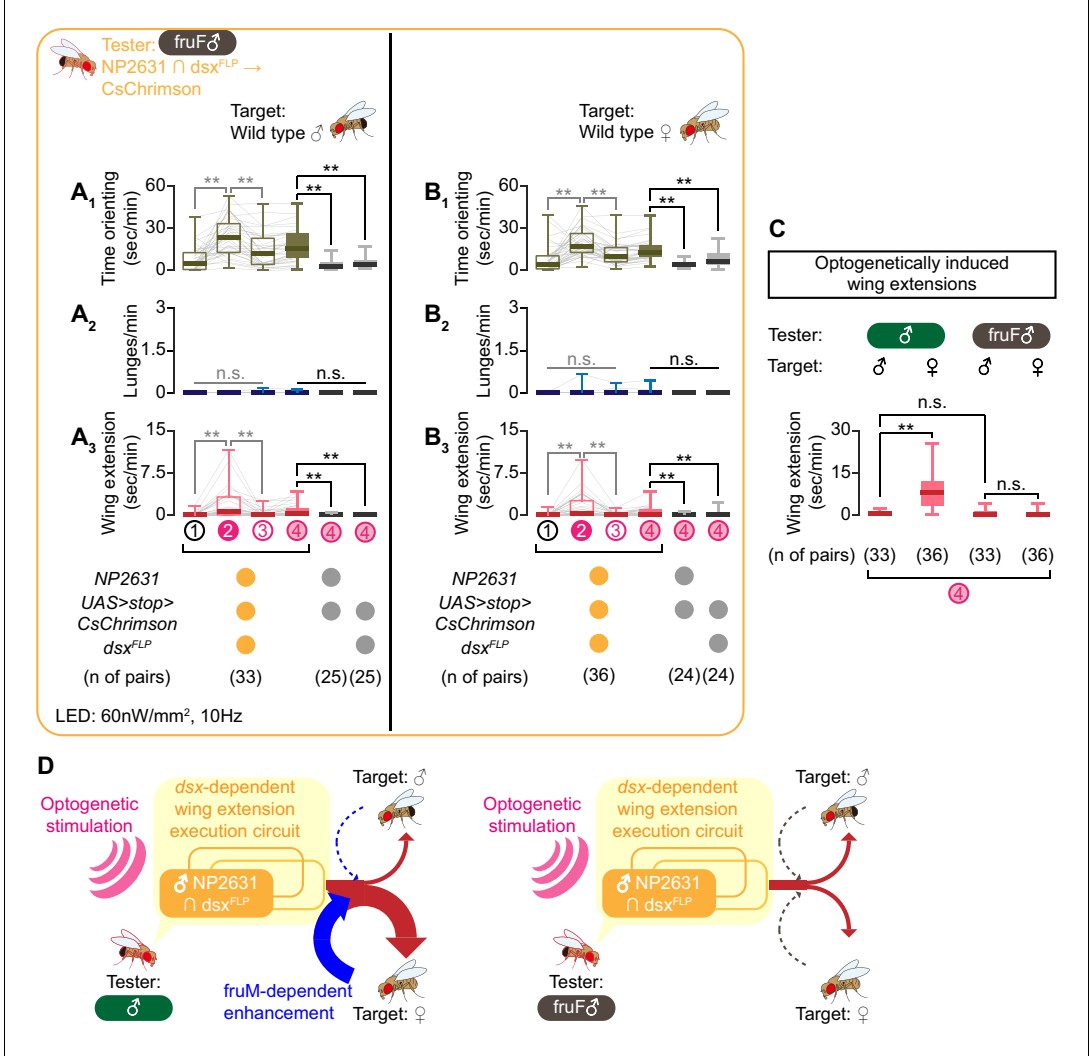

**Figure 4.** NP2631 ∩ dsx$^{FLP}$ neurons in fruF males promote courtship, but not aggressive, behavior. (**A, B**) Boxplots of time orienting (A$_1$, B$_1$), lunges (A$_2$, B$_2$), and wing extension (A$_3$, B$_3$) by the tester flies during the time windows 1–4 (see **Figure 1D**). Testers' genotypes and pair numbers are indicated below the plots. Gray lines represent single testers. Target flies are either group-housed wild-type males (**A**) or mated females (**B**). In gray: **p<0.01, n.s. p>0.05 (Kruskal-Wallis one-way ANOVA and post-hoc Wilcoxon signed rank test). In black: **p<0.01, n.s. p>0.05 (Kruskal-Wallis one-way ANOVA and post-hoc Mann-Whitney U-test). (**C**) Comparison of wing extension performed by flies that express CsChrimson:tdTomato under the control of *NP2631* and *dsx$^{FLP}$* in males (data from **Figure 3—figure supplement 1I**) or in fruF males (data from A, (**B**), during the time window 4. Sex of tester and target flies is indicated above the panels. Number of pairs tested is indicated below the panels. **p<0.01, n.s. p>0.05 (Mann-Whitney U-test). (**D**) Models of the function of NP2631 ∩ dsx$^{FLP}$ neurons in the context of *dsx* and *fru*.

The online version of this article includes the following figure supplement(s) for figure 4:

**Figure supplement 1.** Rasters of behaviors induced by the optogenetic activation of NP2631 ∩ dsx$^{FLP}$ neurons in fruF males.

## Discussion

In this study, we uncovered distinct yet cooperative roles of *dsx* and *fru* on male-type social behaviors through a specific subset of P1/pC1 neurons. For courtship behaviors, we found that NP2631 ∩ dsx$^{FLP}$ neurons are specified in a *fru*-independent manner, and in males, their capacity to generate courtship behaviors does not require fruM. However, activation of NP2631 ∩ dsx$^{FLP}$ neurons in fruF males failed to increase courtship selectively toward female targets. These results suggest that *dsx* plays a major role in establishing a neuronal circuit that enables the male flies to execute courtship behavior, whereas *fru* is critical for enhancing courtship behavior toward females, likely through proper recognition of target sex. The fact that the sex of the target flies influences the function of P1/pC1 subsets implies that information about target sex can modulate the neural circuit units

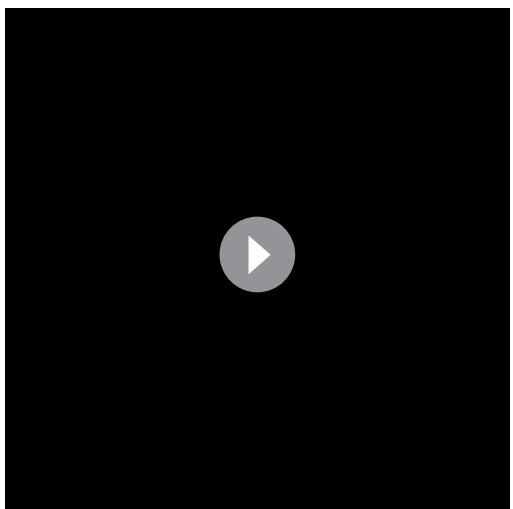

**Video 6.** Representative behavior of a fruF male tester fly that expresses. CsChrimson:tdTomato under the control of *NP2631* and *dsx*$^{FLP}$ toward a wild-type male (Part 1) or a wild-type female (Part 2) target fly, at the onset and offset of LED stimulation.

https://elifesciences.org/articles/52701#video6

downstream of these neurons, and encourages us to revise the linear circuit model for sexually dimorphic social behaviors. In contrast, the complete specification and courtship-promoting functions of P1$^a$ neurons require both *dsx* and *fru*, revealing genetic and functional heterogeneity within P1/pC1 neurons. Lastly, NP2631 $\cap$ dsx$^{FLP}$ and P1$^a$ neurons require a fruM-dependent mechanism to promote male-type aggressive behavior. This suggests that neither of these neurons are part of the execution mechanism for male-type aggressive behavior, and that the genetic mechanisms specifying execution components for courtship and aggressive behaviors are different.

## Sex of the target animals is an important biological variable

Electrical stimulation of various parts of the brain has been known to elicit complex behaviors, including social behaviors, for almost a century (*Koolhaas, 1978*; *Miller, 1957*). Recent technological advances have allowed researchers to identify specific, genetically labeled populations of neurons that can induce mating and aggressive behaviors upon acute optogenetic stimulation in both mice and in flies (*Anderson, 2016*; *Bayless and Shah, 2016*; *Chen and Hong, 2018*; *Li and Dulac, 2018*), even toward suboptimal targets (such as inanimate objects) (*Asahina et al., 2014*; *Duistermars et al., 2018*; *Hong et al., 2014*; *Lin et al., 2011*; *Pan et al., 2012*). These findings seem consistent with the idea that neuronal activation can override most contexts and generate specific behaviors depending on the identity of stimulated cells. However, interactions with a target animal can transmit important information which a tester animal may use to choose appropriate behaviors (*Chen and Hong, 2018*). In fact, attacks triggered by optogenetic stimulation of ventrolateral hypothalamus (VMH) in male mice tend to last longer toward castrated males than toward female targets (*Lin et al., 2011*), and chemogenetic activation of progesterone receptor-expressing VMH neurons appears to induce more attacks toward male than toward female targets (*Yang et al., 2017*). While effects on target sex are not consistently documented, our results and above-mentioned previous observations in mice show that the target sex has a significant impact on behavioral choice even for optogenetically induced social behaviors. These results suggest that sensory or behavioral feedback from target animals can impact the operation of what may appear to be an 'execution mechanism' for a given behavior.

Identification of neural sites where the information about the target sex is integrated with the activity of both NP2631 $\cap$ dsx$^{FLP}$ and P1$^a$ neurons will be an important step in understanding how such context cues modulate ongoing neural activity and, ultimately, behavioral outcome. While a 'command'-like center that irreversibly executes courtship or aggressive behaviors, like recently characterized egg-laying controlling neurons (*Wang et al., 2020*), may exist, it is also possible that information about target sex (and its behavioral response) can be injected at multiple levels of a neural circuit, thereby ensuring the target sex-specific execution of sexually dimorphic social behaviors. This is conceptually analogous to the neural control of fine motions, which can be constantly adjusted by sensory feedback and efference copies all the way down to the motoneuron level (*Azim and Seki, 2019*).

Recently, the importance of addressing sex as a biological variable has been widely recognized (*Klein et al., 2015*; *McCarthy et al., 2017*). In the context of social behaviors, this variable in the tester animals can be critical for uncovering the underlying neural mechanisms.

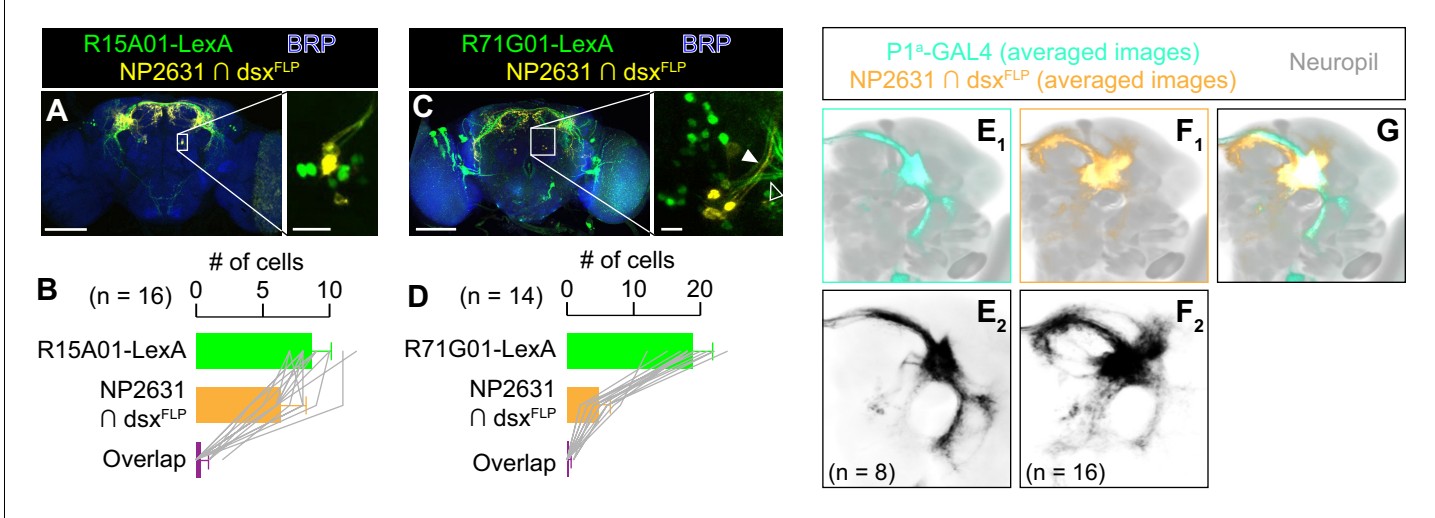

**Figure 5.** P1$^a$ neurons are distinct from NP2631 $\bigcap$ dsx$^{FLP}$ neurons. (**A, C**) Expression of CsChrimson:tdTomato under the control of *NP2631* and *dsx$^{FLP}$* (yellow), GCaMP6f under the control of *R15A01-LexA* (green in **A**) or *R71G01-LexA* (green in **C**), and a neuropil marker BRP (blue) in a representative male brain are visualized by immunohistochemistry. Scale bar: 100 µm. An inset represents a magnified view of the posterior cell body cluster in the white rectangle (scale bar: 10 µm). An open arrowhead in **C**) indicates a neural tract from LexA-expressing neurons that appear distinct from the tract from neurons labeled by *NP2631* and *dsx$^{FLP}$* (white arrowhead). (**B, D**) Mean number of cell bodies per hemibrain with immunohistochemical signal by anti-DsRed antibody (orange), anti-GFP antibody (green), and both antibodies (purple) in brains of the genotype represented in A (**B**) or in C (**D**). In D, all LexA-expressing neurons located near *NP2631* and *dsx$^{FLP}$* neurons are included, although some may belong to different neuronal clusters. Error bars: S.D. Gray lines represent single hemibrains. (**E–G**) Z-projection of the registered and averaged images of CsChrimson:tdTomato expression under the control of *P1$^a$-GAL4* (cyan in E$_1$, G, black in E$_2$), and *NP2631* and *dsx$^{FLP}$* (orange in F$_1$, G, black in F$_2$; duplication from *Figure 3H*) neurons. A part of the standard *Drosophila* brain is shown in gray. Number of hemibrains used are indicated in E$_2$ and F$_2$.

The online version of this article includes the following figure supplement(s) for figure 5:

**Figure supplement 1.** Detailed characterization of P1$^a$ neurons.
**Figure supplement 2.** Detailed characterization of P1$^a$ $\bigcap$ fru$^{FLP}$ neurons.
**Figure supplement 3.** P1$^a$ and NP2631 $\bigcap$ fru$^{FLP}$ neurons do not overlap.

## Organizational function of *dsx* on *Drosophila* courtship behavior

The functional segregation of *dsx* and *fru* that we observed can be considered analogous to the organizational and activation functions of sex hormones in mammals (*McCarthy, 2008*). Differential exposure to gonadal steroid hormones, mostly through estrogen receptors (*Wu et al., 2009*), specifies neural circuits that are necessary for sex-specific reproductive behaviors, whereas hormonal surges in the adult stage (such as testosterone (*Juntti et al., 2010*) or progesterone *Dey et al., 2015*; *Inoue et al., 2019*) orchestrate activation of sex-specific behaviors. We postulate that *dsx* has an organizational function for the courtship execution circuit, whereas *fru* is important for the appropriate activation of the circuit.

Our results do not mean that *fru* is not necessary for the establishment of *all* neuronal components involved in courtship (see below). Nonetheless, our result suggests that the wing extension execution circuit that connects NP2631 $\bigcap$ dsx$^{FLP}$ neurons and relevant motoneurons is specified even in the absence of fruM (*Pan et al., 2011*), which is consistent with previous observations that fruF males are capable of expressing at least a part of courtship behavior (*Demir and Dickson, 2005*; *Hall, 1978*; *Villella et al., 1997*). While a specification role

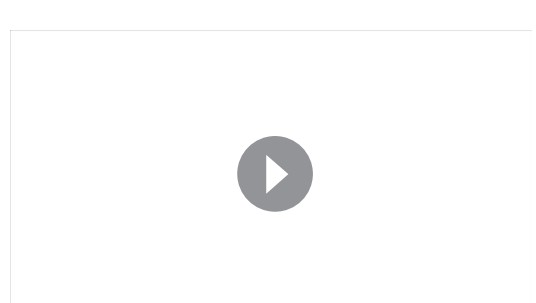

**Video 7.** 3D-rendered average image of P1$^a$ neurons (cyan) and NP2631 $\bigcap$ dsx$^{FLP}$ neurons (orange) in male. Gray represents a standard unisex *Drosophila* brain (*Bogovic et al., 2018*).
https://elifesciences.org/articles/52701#video7

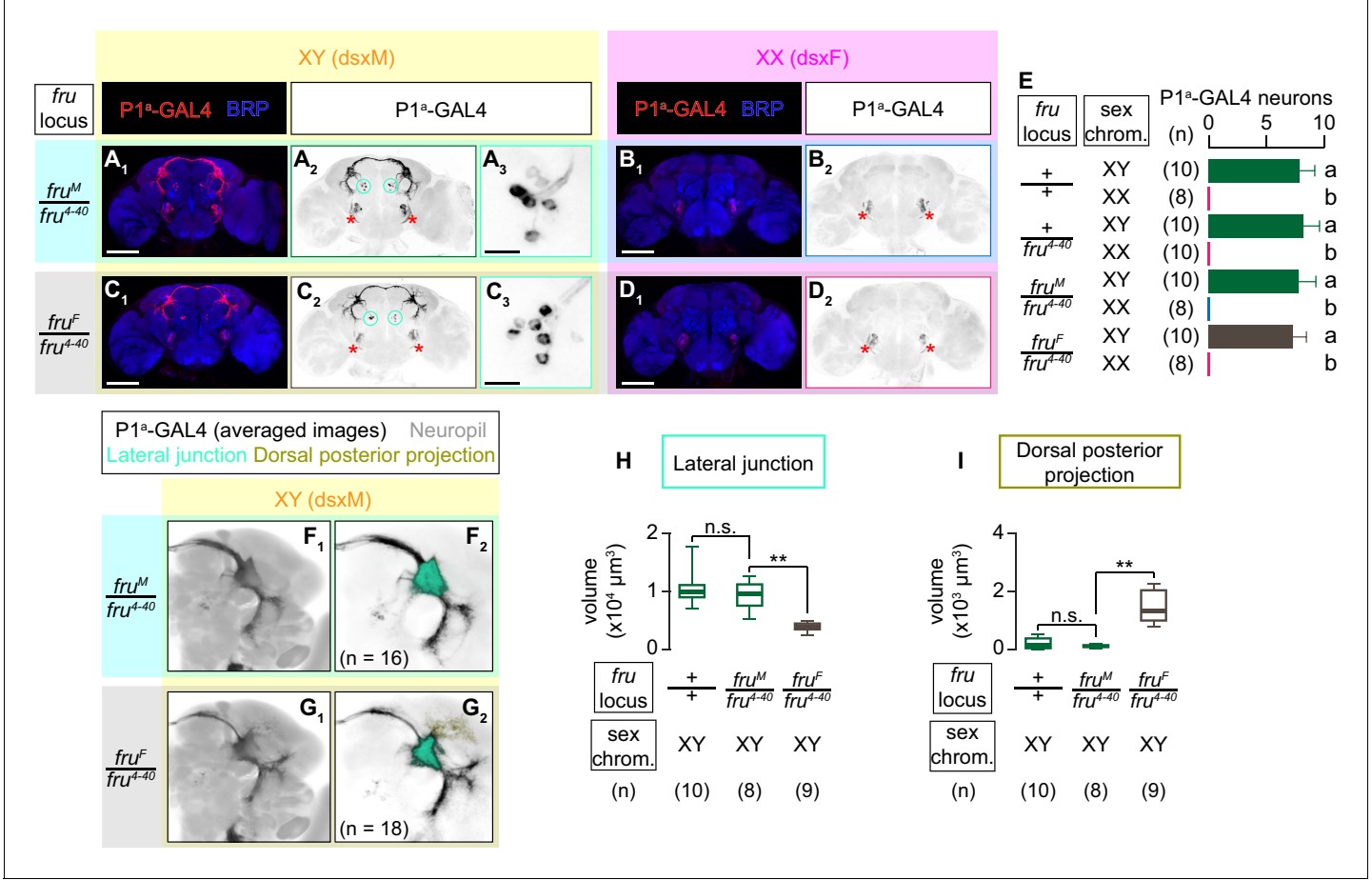

**Figure 6.** Both *dsx* and *fru* specify the sexual dimorphism of P1$^a$ neurons. (A–D) Expression of CsChrimson:tdTomato under the control of *P1$^a$-GAL4* (red in A$_1$–D$_1$, black in A$_{2,3}$-D$_{2,3}$) in brains of a male (A), fruM female (B), fruF male (C), and female (D) is visualized together with a neuropil marker BRP (blue) by immunohistochemistry. Circle: soma (right cluster is enlarged in A$_3$ and C$_3$), red asterisk: sex-invariant background labeling (see Materials and methods for details). Scale bar: 100 μm (A$_1$–D$_1$), 10 μm (A$_3$, C$_3$). (E) Mean number of cell bodies per hemibrain visualized by anti-DsRed antibody in each genotype represented in A–D) and *Figure 6—figure supplement 1A–D*. (F, G) Z-projection of the registered and averaged images of CsChrimson:tdTomato expression under the control of *P1$^a$-GAL4* in male (F) and fruF male (G). A part of the standard *Drosophila* brain is shown in gray in F$_2$, G$_2$. Number of hemibrains used are indicated in F$_2$) and G$_2$). Lateral junction (cyan) and dorsal posterior projection (yellow) are segmented and overlaid in F$_2$) and G$_2$. H, I. Boxplot of volumes of lateral junction (H) and dorsal posterior projection (I). Genotypes and number of hemibrains are indicated below the plot. **p<0.01, n.s. p>0.05 (Kruskal-Wallis one-way ANOVA and post-hoc Mann-Whitney U-test).

The online version of this article includes the following figure supplement(s) for figure 6:

**Figure supplement 1.** Heterozygosity of the *fru$^{4-40}$* allele leaves the neuroanatomy and function of P1$^a$ neurons largely unaltered.

**Figure supplement 2.** P1$^a$ neurons in fruF males are defective in promoting both courtship and aggressive behavior.

for *dsx* on P1/pC1 neurons has been previously reported (*Kimura et al., 2008*; *Pan et al., 2011*; *Rideout et al., 2010*; *Sanders and Arbeitman, 2008*; *von Philipsborn et al., 2014*), our study showed for the first time the behavioral role of a specific P1/pC1 subset (NP2631 ∩ dsx$^{FLP}$ neurons) in fruF males. *dsx* is important for the specification of a few other behaviorally relevant sexual dimorphisms in the *Drosophila* nervous system. For instance, the sexually dimorphic axon development of leg gustatory receptor neurons, which includes aphrodisiac pheromone sensors (*Lu et al., 2012*; *Starostina et al., 2012*; *Thistle et al., 2012*; *Toda et al., 2012*), requires *dsx* function (*Mellert et al., 2010*; *Mellert et al., 2012*). The neural connectivity and function of TN1 neurons (*Rideout et al., 2010*; *Robinett et al., 2010*; *Shirangi et al., 2016*), which are pre-motor neurons important for the production of pulse song, are also specified by *dsx* (*Shirangi et al., 2016*). Several classes of abdominal ganglia neurons involved in male copulation also express *dsx* (*Crickmore and Vosshall, 2013*; *Pavlou et al., 2016*). Although relatively few in number, these examples display the

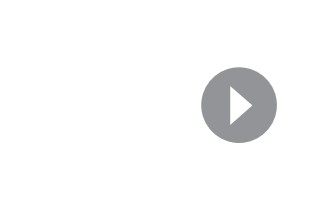

**Video 8.** 3D-rendered average image of P1ᵃ neurons in male (green) and in fruF male (yellow). Gray represents a standard unisex *Drosophila* brain (*Bogovic et al., 2018*).
https://elifesciences.org/articles/52701#video8

importance of *dsx* in key neuronal populations for organizing circuit components that are essential for the execution of courtship behaviors. It is noteworthy that *dsx* is involved in sex-determination across a variety of animal phyla (*Kopp, 2012*; *Matson and Zarkower, 2012*), whereas *fru*'s role in sex-determination seems confined to insects (*Gailey et al., 2006*). This suggests that *dsx* may be evolutionarily more ancient in the context of sex-determination than *fru*, which can account for its dominance over *fru* when specifying sexually dimorphic neurons that co-express *dsx* and *fru*.

## Function of *fru* in activating courtship toward female targets

Our proposal that fruM may be important for enhancing courtship behavior specifically towards females is consistent with the fact that many characterized *fru*-expressing neurons are involved in processing sex- and species-specific sensory cues (*Clowney et al., 2015*; *Fan et al., 2013*; *Kallman et al., 2015*; *Kohl et al., 2013*; *Kurtovic et al., 2007*; *Lin et al., 2016*; *Lu et al., 2012*; *Ribeiro et al., 2018*; *Ruta et al., 2010*; *Starostina et al., 2012*; *Stockinger et al., 2005*; *Thistle et al., 2012*; *Toda et al., 2012*; *Yu et al., 2010*; *Zhou et al., 2015*). Namely, P1ᵃ neurons, as well as more broadly defined P1/pC1 neurons accessed by different genetic reagents, are known to respond to sex-specific chemical cues (*Clowney et al., 2015*; *Kallman et al., 2015*; *Kohatsu et al., 2011*), underscoring their critical role in sensory integration for courtship (*Auer and Benton, 2016*; *Ellendersen and von Philipsborn, 2017*). fruM can play the 'activation' role for courtship by establishing sensory circuits that transmit sex-specific sensory information to P1/pC1 neurons (*Ito et al., 2012*; *Ito et al., 2016*; *Kimura et al., 2005*; *von Philipsborn et al., 2014*), or by enabling P1/pC1 neurons to properly integrate and transform such neural inputs. Neuroanatomical defects of P1ᵃ neurons in fruF males could disrupt either process.

Gain control of sex-specific sensory cues can be one neuronal mechanism for the 'activation' function, but courtship behavior can be enhanced in other ways as well. For instance, behavioral persistence or context-dependent intensity adjustment (*Clemens et al., 2018*; *Coen et al., 2016*; *Grosjean et al., 2011*; *Keleman et al., 2012*; *Zhang et al., 2016*) can result in an increase of the overall courtship vigor. Recently, a new class of *fru*-expressing neurons downstream of P1ᵃ neurons has been found to mediate the persistence of courtship behavior triggered by P1ᵃ neuronal activation (*Jung et al., 2020*). Even if fruM is not absolutely necessary for the formation of the minimal wing extension execution circuit, it can have a significant impact on the generation of effective wing extension toward female target flies (*Fan et al., 2013*; *Pan and Baker, 2014*; *Pan et al., 2011*).

While we conclude that the role of *fru* is not necessarily to specify the execution mechanism for courtship behavior, fruM females can still perform wing extensions (*Demir and Dickson, 2005*; *Rideout et al., 2007*). Moreover, artificial stimulation of either *fru*- (*Clyne and Miesenböck, 2008*; *Pan et al., 2011*) or *dsx*- (*Rezával et al., 2016*) expressing neurons in females can elicit wing extensions, suggesting that the residual execution mechanism for at least a part of courtship behavior may be specified in a sex-invariant manner. The presence of a latent mating execution circuit in female brains is also suggested in mice (*Kimchi et al., 2007*; *Li and Dulac, 2018*). Because the courtship songs produced by females or fruM females are defective (*Pan et al., 2011*; *Rezával et al., 2016*; *Rideout et al., 2007*), male-type splicing of *dsx* nonetheless seems to be instrumental in organizing the proper execution mechanism for *Drosophila* courtship behavior.

In striking contrast to wing extensions, we found that activation of neither NP2631 ∩ dsx^FLP nor P1ᵃ neurons in fruF males induced lunges. This result points to the existence of a fruM-dependent execution mechanism for male-type aggressive behaviors (*Vrontou et al., 2006*), likely downstream of these neurons. In *Wohl et al. (2020)*, we found that at least one group of fruM-dependent neurons can promote male-type aggressive behaviors independent of *dsx*. Therefore, a separation of the courtship execution mechanism and the aggression execution mechanism by two sex-determining genes is likely accomplished by a partial separation of underlying neural circuits. The aggression-

promoting function of NP2631 ∩ dsx[FLP] and P1[a] neurons likely reflects their roles to coordinate aggression and courtship depending on internal and external conditions, instead of a simple decision switch that triggers fixed types of behavior.

## An organismal sex and a cellular sex

Both *dsx* and *fru* encode transcription factors. The sexually dimorphic morphology and wiring specificity of many *fru*-expressing neurons are determined in a cell-autonomous manner (*Kimura et al., 2008*; *Kohl et al., 2013*), suggesting that *dsx* and *fru* define *Drosophila* sex at a cellular level through regulation of a specific set of target genes (*Dalton et al., 2013*; *Ito et al., 2012*; *Neville et al., 2014*; *Vernes, 2015*). Mammalian sex hormones ultimately exert their effects through nuclear steroid receptors, which serve as transcription factors. Thus, both in flies and in mammals, organismal sex can be regarded as a collective phenotype of genetic 'sexes' that can be reduced down to the cellular level (*Robinett et al., 2010*; *Williams and Carroll, 2009*). To understand how sex at the neuronal level influences sexually dimorphic behaviors, cell-type specific manipulation of sex-determining genes is required. Our current study focused on neural functions in a whole animal mutant, which prevents us from addressing the role of either *dsx* or *fru* specifically within NP2631 ∩ dsx[FLP] or P1[a] neurons. For example, we do not know whether NP2631 ∩ dsx[FLP] neurons in fruF males failed to enhance courtship behavior toward female targets because of the absence of fruM within this population, or because of the lack of fruM in other neuronal populations, or both. In addition, our four genotypes approach does not address if it is the presence of dsxM or the absence of dsxF that is important for the specification of male-type NP2631 ∩ dsx[FLP] neurons or P1[a] neurons.

It is important to note that sex specification is a developmental process of transformation. Both at genetic and organismal levels, one sex is not a loss-of-function mutant of the other. Loss-of-function manipulations at the cellular level, by cell type-specific RNA interference (*Dietzl et al., 2007*; *Ni et al., 2009*) or CRISPR interference (*Qi et al., 2013*)-based approaches, may show that either *dsx* or *fru* is necessary for the proper development or function of the given neurons, but may be insufficient to illuminate the genetic origin of the sex-specific transformation at the cellular level. In addition, temporally and spatially precise manipulation of genes during development remains difficult. This can create a difficulty interpreting the effects of either knock-down or over-expression of sex determining genes, which are dynamically regulated from early developmental stages (*Baker and Ridge, 1980*; *Lee et al., 2000*; *Lee et al., 2002*; *Mellert et al., 2012*; *Sanders and Arbeitman, 2008*). Creation of neuronal mutant clones (*Lee and Luo, 1999*) may circumvent this problem, but the *tra* mutation (*Kimura et al., 2008*; *Kohl et al., 2013*), which has been previously used to convert a 'neuronal sex', cannot dissociate the roles of *dsx* and *fru* (see *Figure 3A*).

Faced with these often overlooked limitations of cell-type specific gene manipulations, it would be informative to characterize what types of transformations are observed in mutants of sex-specific splicing at an organismal level, as in this and other studies (*Datta et al., 2008*; *Kimura et al., 2008*; *Kohl et al., 2013*; *von Philipsborn et al., 2014*; *Zhou et al., 2015*). Although a constitutive mutants have above-mentioned limitations, they nonetheless establish fundamental functional differences among sex-determining genes, as well as benchmarks for the efficacy for cell-specific manipulations techniques. Although clearly out of the scope of the current study, electron microscopy-based connectome reconstructions of fruF male and fruM female brains (*Zheng et al., 2018*) could provide useful information for understanding the transformative nature of sex specification in the brain.

## Uncovering functional heterogeneity of social behavior-controlling neurons

Lastly, our serendipitous finding that NP2631 ∩ dsx[FLP] and P1[a] neurons contain genetically and functionally distinct populations underscores the importance of characterizing neuronal cell types in greater detail. How to determine cell types remains a challenge in neuroscience, but genetic access to a finely defined population of neurons even within what is considered as a single class of neurons can be the key to understand how a neural circuit generates complex behaviors such as social behaviors (*Luo et al., 2018*).

In the posterior part of male brains, 'P1' neurons, as defined by *fru*-expressing cluster (*Kimura et al., 2008*; *Lee et al., 2000*), and pC1 neurons, as defined by *dsx*-expressing cluster (*Lee et al., 2002*; *Rideout et al., 2010*), extensively overlap (*Rideout et al., 2007*; *Rideout et al.,*

*2010*; *Sanders and Arbeitman, 2008*; *Zhou et al., 2014*) This raises a question about the distinction between 'P1' and 'pC1' neurons (*Asahina, 2018*). Furthermore, recent single cell level analyses of the neurons that belong to the male 'P1' cluster or 'pC1' cluster revealed surprising neuroanatomical and functional diversity (*Costa et al., 2016*; *Zhang et al., 2018*), raising a possibility that P1/pC1 neurons may be functionally heterogeneous as well (*Asahina, 2018*).

Surprisingly, we found that behaviorally relevant NP2631 ∩ dsx$^{FLP}$ and P1$^a$ neurons, as well as NP2631 ∩ fru$^{FLP}$ and P1$^a$ neurons, seldom overlap. Optogenetic stimulation of NP2631 ∩ dsx$^{FLP}$ and P1$^a$ neurons triggers social behaviors in temporally distinct manners. Moreover, *fru* has a different impact on the specification and function of these two neuron groups, suggesting that little overlap of NP2631 ∩ dsx$^{FLP}$ and P1$^a$ neurons does not necessarily reflect arbitrary labeling bias within a single homogeneous neuronal population by different genetic reagents. Instead, these observations support the idea that of P1/pC1 neurons consist of functionally diverse subtypes.

We acknowledge that the genetic reagents used in our study are likely insufficient to resolve the possible heterogeneity within either NP2631 ∩ dsx$^{FLP}$ or P1$^a$ neurons. Differential expression patterns of FruM proteins within both clusters (*Figure 2—figure supplement 2C*, *Figure 5—figure supplement 1D*) alone suggest that such heterogeneity almost certainly exists. Recent advances in whole-brain neural reconstruction using electron microscopy images (*Zheng et al., 2018*) will provide a foundation for precise characterization of *Drosophila* neurons, as has been recently used for the female-type 'pC1' cluster (*Deutsch et al., 2020*; *Wang et al., 2020*). A large number of 'split-GAL4' collections (*Jenett et al., 2012*; *Kvon et al., 2014*; *Luan et al., 2006*) will allow universal access to the specific subpopulations. These types of tools will facilitate cross-study comparisons of neuroanatomical and behavioral data, and will serve as a catalyst to understand the logic of neural control of behavior in general. With the advance of single cell-level genetic and epigenetic profiling techniques, the importance of precisely characterizing the targeted neuronal types will only grow not only in *Drosophila*, but in every model organism. Reproducible access to each neuronal type can uncover functional units for a given behavior at even finer detail (*Robie et al., 2017*), which will be fundamental for deconstructing the dynamics of neural circuits that are responsible for generating social behaviors in a context-dependent manner. Such knowledge will be also critical for establishing theoretical models that account for brain operations (*Kingsbury et al., 2019*; *Zhang and Yartsev, 2019*) and population-level dynamics of animals (*Ramdya et al., 2017*) engaging in social interactions.

## Materials and methods

See *Supplementary file 1* for details of reagents used in this study.

### Fly strains

See *Table 1* for the complete genotypes of *Drosophila* strains used in each figure panel. *NP2631* (*Yu et al., 2010*) is a gift from Daisuke Yamamoto (Tohoku University). *P1$^a$-GAL4* (*R15A01-p65AD: Zp* (in attP40) (RRID:BDSC_68837); *R71G01-Zp:GAL4DBD* (in attP2) (RRID:BDSC_69507)) (*Hoopfer et al., 2015*; *Inagaki et al., 2014*) and *R15A01-LexA* (in attP2) (*Hoopfer et al., 2015*) were gifts from David Anderson (California Institute of Technology). *20XUAS-IVS-Syn21-CsChrimson: tdTomato* (in VK00022), *20XUAS > myr:TopHAT2 >CsChrimson:tdTomato* (in VK00022 and VK00005) (*Duistermars et al., 2018*; *Watanabe et al., 2017*) and *13XLexAop2-IVS-Syn21-GCaMP6f* (codon-optimized)-*p10* (in su(Hw)attP5) were created by Barret Pfeiffer in the lab of Gerald Rubin (HHMI Janelia Research Campus) and kindly shared by David Anderson. *fru$^M$* (RRID:BDSC_66874), *fru$^F$* (RRID:BDSC_66873) (*Demir and Dickson, 2005*), and *fru$^{FLP}$* (RRID:BDSC_66870) (*Yu et al., 2010*) flies are gifts from Barry Dickson (HHMI Janelia Research Campus); *dsx$^{GAL4}$* (*Rideout et al., 2010*) and *dsx$^{FLP}$* (*Rezával et al., 2014*) are gifts from Stephen Goodwin (University of Oxford); *fru$^{P1.LexA}$* (RRID:BDSC_66698) (*Mellert et al., 2010*) is a gift from Bruce Baker (HHMI Janelia Research Campus). *fru$^{4-40}$* (RRID:BDSC_66692), and *8XLexAop2-GAL80* (in attP40) (RRID:BDSC_32214) flies were obtained from Bloomington *Drosophila* Resource Center in the University of Indiana.

*R71G01-LexA* (in attP2) was created by targeting the GMR71G01-LexA plasmid construct (in pCR8/GW/TOPO backbone: gift from Gerald Rubin, HHMI Janelia Research Campus) into the P {CaryP}attP2 landing site. The plasmid was injected into embryos of the *nos-phiC31, y$^1$, sc$^1$, v$^1$,*

**Table 1.** Complete genotypes of *Drosophila* strains used in this study.

| Figure | Panel | Abbreviated genotype | COMPLETE GENOTYPE ('Y' represents the Y chromosome) |
|---|---|---|---|
| *Figure 1* | A, B | dsx$^{GAL4}$ ∩ fru$^{FLP}$ ♂ | *w/Y; 20XUAS > myr:TopHAT2 > CsChrimson: tdTomato in VK00022/+; dsx$^{GAL4}$/fru$^{FLP}$* |
| *Figure 1—figure supplement 1* | A | | |
| *Video 1* | | | |
| *Figure 1* | B | dsx$^{GAL4}$ ∩ fru$^{FLP}$ ♀ | *w/w; 20XUAS > myr:TopHAT2 > CsChrimson: tdTomato in VK00022/+; dsx$^{GAL4}$/fru$^{FLP}$* |
| *Figure 1—figure supplement 1* | B | | |
| *Figure 1* | E-I | *dsx$^{GAL4}$, UAS > stop > CsChrimson, fru$^{FLP}$* | *w/Y; 20XUAS > myr:TopHAT2 > CsChrimson: tdTomato in VK00022/+; dsx$^{GAL4}$/fru$^{FLP}$* |
| *Figure 1—figure supplement 1* | C, D | | |
| *Figure 1* | E-H | *dsx$^{GAL4}$, UAS > stop > CsChrimson* | *w/Y; 20XUAS > myr:TopHAT2 > CsChrimson: tdTomato in VK00022/+; dsx$^{GAL4}$/+* |
| *Figure 1—figure supplement 1* | C, D | | |
| *Figure 1* | E-H | *UAS > stop > CsChrimson, fru$^{FLP}$* | *w/Y; 20XUAS > myr:TopHAT2 > CsChrimson: tdTomato in VK00022/+; fru$^{FLP}$/+* |
| *Figure 1—figure supplement 1* | C, D | | |
| *Figure 1—figure supplement 2* | B | Canton-S ♂ | *+/Y; +/+; +/+ (Canton-S)* |
| *Figure 1—figure supplement 2* | B | Canton-S ♀ | *+/+; +/+; +/+ (Canton-S)* |
| *Figure 2* | A, B | *NP2631, UAS > stop > CsChrimson, dsx$^{FLP}$* | *w/Y; NP2631/20XUAS > myr:TopHAT2 > CsChrimson: tdTomato in VK00022; dsx$^{FLP}$, fru$^{4-40}$/+* |
| *Figure 2—figure supplement 1* | A-C | | |
| *Figure 2—figure supplement 3* | A, B | | |
| *Video 2* | | | |
| *Figure 2* | A, B | *NP2631, UAS > stop > CsChrimson* | *w/Y; NP2631/20XUAS > myr:TopHAT2 > CsChrimson: tdTomato in VK00022; fru$^{4-40}$/+* |
| *Figure 2—figure supplement 3* | A, B | | |
| *Figure 2* | A, B | *UAS > stop > CsChrimson, dsx$^{FLP}$* | *w/Y; +/20XUAS > myr:TopHAT2 > CsChrimson: tdTomato in VK00022; dsx$^{FLP}$, fru$^{4-40}$/+* |
| *Figure 2—figure supplement 3* | A, B | | |
| *Figure 2* | C | *NP2631, UAS > stop > CsChrimson, dsx$^{FLP}$, fru$^{P1. LexA}$, LexAop2-GAL80* | *w/Y; NP2631/8XLexAop2-GAL80 in attP40; 20XUAS > myr:TopHAT2 > CsChrimson: tdTomato in VK00005, dsx$^{FLP}$/fru$^{P1. LexA}$* |
| *Figure 2* | C | *NP2631, UAS > stop > CsChrimson, dsx$^{FLP}$, LexAop2-GAL80* | *w/Y; NP2631/8XLexAop2-GAL80 in attP40; 20XUAS > myr:TopHAT2 > CsChrimson: tdTomato in VK00005, dsx$^{FLP}$/+* |
| *Figure 2* | C | *NP2631, UAS > stop > CsChrimson, dsx$^{FLP}$, fru$^{P1. LexA}$* | *w/Y; NP2631/+; 20XUAS > myr:TopHAT2 > CsChrimson: tdTomato in VK00005, dsx$^{FLP}$/fru$^{P1. LexA}$* |
| *Figure 2—figure supplement 1* | A-C | *NP2631, UAS > stop > CsChrimson, dsx$^{FLP}$* | *w/Y; NP2631/20XUAS > myr:TopHAT2 > CsChrimson: tdTomato in VK00022; dsx$^{FLP}$, fru$^{4-40}$/+* |
| *Figure 2—figure supplement 2* | A-D | NP2631 ∩ dsx$^{FLP}$ | *w/Y; NP2631/20XUAS > myr:TopHAT2 > CsChrimson: tdTomato in VK00022; dsx$^{FLP}$/+* |
| *Figure 2—figure supplement 2* | E, H | *NP2631, UAS > stop > CsChrimson (VK00005), dsx$^{FLP}$, fru$^{P1. LexA}$, LexAop2-GAL80* | *w/Y; NP2631/8XLexAop2-GAL80 in attP40; 20XUAS > myr:TopHAT2 > CsChrimson: tdTomato in VK00005, dsx$^{FLP}$/fru$^{P1. LexA}$* |
| *Figure 2—figure supplement 2* | F, H | *NP2631, UAS > stop > CsChrimson (VK00005), dsx$^{FLP}$, LexAop2-GAL80* | *w/Y; NP2631/8XLexAop2-GAL80 in attP40; 20XUAS > myr:TopHAT2 > CsChrimson: tdTomato in VK00005, dsx$^{FLP}$/+* |
| *Figure 2—figure supplement 2* | G, H | *NP2631, UAS > stop > CsChrimson (VK00005), dsx$^{FLP}$, fru$^{P1. LexA}$* | *w/Y; NP2631/+; 20XUAS > myr:TopHAT2 > CsChrimson: tdTomato in VK00005, dsx$^{FLP}$/fru$^{P1. LexA}$* |
| *Figure 3* | C, G, H, L-N | NP2631 ∩ dsx$^{FLP}$, XY, *fru* locus: *fru$^{M}$/fru$^{4-40}$* | *w/Y; NP2631/20XUAS > myr:TopHAT2 > CsChrimson: tdTomato in VK00022; dsx$^{FLP}$, fru$^{4-40}$/fru$^{M}$* |
| *Figure 3—figure supplement 1* | F | | |
| *Video 3*, 4 | | | |
| *Figure 3* | D, G, I, L-N | NP2631 ∩ dsx$^{FLP}$, XX, *fru* locus: *fru$^{M}$/fru$^{4-40}$* | *w/w; NP2631/20XUAS > myr:TopHAT2 > CsChrimson: tdTomato in VK00022; dsx$^{FLP}$, fru$^{4-40}$/fru$^{M}$* |
| *Video 5* | | | |
| *Figure 3* | E, G, J, L-N | NP2631 ∩ dsx$^{FLP}$, XY, *fru* locus: *fru$^{F}$/fru$^{4-40}$* | *w/Y; NP2631/20XUAS > myr:TopHAT2 > CsChrimson: tdTomato in VK00022; dsx$^{FLP}$, fru$^{4-40}$/fru$^{F}$* |
| *Video 4* | | | |

*Table 1 continued on next page*

*Table 1 continued*

| Figure | Panel | Abbreviated genotype | COMPLETE GENOTYPE ('Y' represents the Y chromosome) |
|---|---|---|---|
| *Figure 3* | F, G, K-N | NP2631 ∩ dsx$^{FLP}$, XX, *fru* locus: *fru$^F$*/*fru$^{4-40}$* | *w/w; NP2631/20XUAS > myr:TopHAT2 > CsChrimson: tdTomato in VK00022; dsx$^{FLP}$, fru$^{4-40}$/fru$^F$* |
| *Figure 3—figure supplement 1* | H | | |
| *Video 3*, 5 | | | |
| *Figure 3* | G | NP2631 ∩ dsx$^{FLP}$, XY, *fru* locus: +/+ | *w/Y; NP2631/20XUAS > myr:TopHAT2 > CsChrimson: tdTomato in VK00022; dsx$^{FLP}$/+* |
| *Figure 3—figure supplement 1* | A, E | | |
| *Figure 3* | G | NP2631 ∩ dsx$^{FLP}$, XX, *fru* locus: +/+ | *w/w; NP2631/20XUAS > myr:TopHAT2 > CsChrimson: tdTomato in VK00022; dsx$^{FLP}$/+* |
| *Figure 3—figure supplement 1* | B, G | | |
| *Figure 3* | G | NP2631 ∩ dsx$^{FLP}$, XY, *fru* locus: +/*fru$^{4-40}$* | *w/Y; NP2631/20XUAS > myr:TopHAT2 > CsChrimson: tdTomato in VK00022; dsx$^{FLP}$, fru$^{4-40}$/+* |
| *Figure 3—figure supplement 1* | C | | |
| *Figure 3* | G | NP2631 ∩ dsx$^{FLP}$, XX, *fru* locus: +/*fru$^{4-40}$* | *w/w; NP2631/20XUAS > myr:TopHAT2 > CsChrimson: tdTomato in VK00022; dsx$^{FLP}$, fru$^{4-40}$/+* |
| *Figure 3—figure supplement 1* | D | | |
| *Figure 3—figure supplement 1* | I | *NP2631, UAS > stop > CsChrimson, dsx$^{FLP}$, fru* locus: +/+ | *w/Y; NP2631/20XUAS > myr:TopHAT2 > CsChrimson: tdTomato in VK00022; dsx$^{FLP}$/+* |
| *Figure 3—figure supplement 1* | I | *NP2631, UAS > stop > CsChrimson, dsx$^{FLP}$, fru* locus: +/*fru$^{4-40}$* | *w/Y; NP2631/20XUAS > myr:TopHAT2 > CsChrimson: tdTomato in VK00022; dsx$^{FLP}$, fru$^{4-40}$/+* |
| *Figure 3—figure supplement 1* | I | *NP2631, UAS > stop > CsChrimson, dsx$^{FLP}$, fru* locus: *fru$^M$*/*fru$^{4-40}$* | *w/Y; NP2631/20XUAS > myr:TopHAT2 > CsChrimson: tdTomato in VK00022; dsx$^{FLP}$, fru$^{4-40}$/fru$^M$* |
| *Figure 4* | A-C | *fruF ♂, NP2631, UAS > stop > CsChrimson, dsx$^{FLP}$* | *w/Y; NP2631/20XUAS > myr:TopHAT2 > CsChrimson: tdTomato in VK00022; dsx$^{FLP}$, fru$^{4-40}$/fru$^F$* |
| *Figure 4—figure supplement 1* | A, B | | |
| *Video 6* | | | |
| *Figure 4* | A, B | *fruF ♂, NP2631, UAS > stop > CsChrimson* | *w/Y; NP2631/20XUAS > myr:TopHAT2 > CsChrimson: tdTomato in VK00022; fru$^{4-40}$/fru$^F$* |
| *Figure 4* | A, B | *fruF ♂, UAS > stop > CsChrimson, dsx$^{FLP}$* | *w/Y; +/20XUAS > myr:TopHAT2 > CsChrimson: tdTomato in VK00022; dsx$^{FLP}$, fru$^{4-40}$/fru$^F$* |
| *Figure 4* | C | *♂, NP2631, UAS > stop > CsChrimson, dsx$^{FLP}$* | *w/Y; NP2631/20XUAS > myr:TopHAT2 > CsChrimson: tdTomato in VK00022; dsx$^{FLP}$, fru$^{4-40}$/fru$^M$* |
| *Figure 5* | A, B | NP2631 ∩ dsx$^{FLP}$, R15A01-LexA | *w/Y; NP2631/13XLexAop2-IVS-GCaMP6f-p10 in su(Hw)attP5; R15A01-LexA in attP2/20XUAS > myr:TopHAT2 > CsChrimson: tdTomato in VK00005, dsx$^{FLP}$* |
| *Figure 5* | C, D | NP2631 ∩ dsx$^{FLP}$, R71G01-LexA | *w/Y; NP2631/13XLexAop2-IVS-GCaMP6f-p10 in su(Hw)attP5; 20XUAS > myr:TopHAT2 > CsChrimson: tdTomato in VK00005, dsx$^{FLP}$/R71G01-LexA in attP2* |
| *Figure 5* | E, G | P1$^a$-GAL4 | *w/Y; R15A01-p65AD:Zp in attP40/20XUAS-IVS-CsChrimson:tdTomato in VK00022; R71G01-Zp:GAL4DBD in attP2/+* |
| *Video 7* | | | |
| *Figure 5* | F, G | NP2631 ∩ dsx$^{FLP}$ | *w/Y; NP2631/20XUAS > myr:TopHAT2 > CsChrimson: tdTomato in VK00022; dsx$^{FLP}$, fru$^{4-40}$/fru$^M$* |
| *Video 7* | | | |
| *Figure 5—figure supplement 1* | A | *P1$^a$-GAL4, UAS-CsChrimson* | *w/Y; R15A01-p65AD:Zp in attP40/ 20XUAS-IVS-CsChrimson:tdTomato in VK00022; R71G01-Zp:GAL4DBD in attP2/+* |
| *Figure 5—figure supplement 1* | A | *P1$^a$-GAL4* | *w/Y; R15A01-p65AD:Zp in attP40/ 20XUAS > myr:TopHAT2 > CsChrimson: tdTomato in VK00022; R71G01-Zp: GAL4DBD in attP2/+* |
| *Figure 5—figure supplement 1* | A | *UAS-CsChrimson* | *w/Y; +/20XUAS-IVS-CsChrimson: tdTomato in VK00022; +/+* |
| *Figure 5—figure supplement 1* | B | *P1$^a$-GAL4, UAS-CsChrimson* | *w/Y; R15A01-p65AD:Zp in attP40/ 20XUAS-IVS-CsChrimson:tdTomato in VK00022; R71G01-Zp:GAL4DBD in attP2, fru$^{4-40}$/fru$^M$* |

Table 1 continued

| Figure | Panel | Abbreviated genotype | COMPLETE GENOTYPE ('Y' represents the Y chromosome) |
|---|---|---|---|
| *Figure 5—figure supplement 1* | B | P1ª-GAL4 | *w/Y; R15A01-p65AD:Zp in attP40/ 10XUAS-IVS-GFP-p10 in VK00022; R71G01-Zp:GAL4DBD in attP2, fru⁴⁻⁴⁰/fruᴹ* |
| *Figure 5—figure supplement 1* | B | UAS-CsChrimson | *w/Y; +/20XUAS-IVS-CsChrimson: tdTomato in VK00022; fru⁴⁻⁴⁰/fruᴹ* |
| *Figure 5—figure supplement 1* | C, D | P1ª-GAL4 | *w/Y; R15A01-p65AD:Zp in attP40/ 20XUAS-IVS-CsChrimson:tdTomato in VK00022; R71G01-Zp:GAL4DBD in attP2/+* |
| *Figure 5—figure supplement 2* | A | P1ª-GAL4, UAS-CsChrimson (VK00022) | *w/Y; R15A01-p65AD:Zp in attP40/ 20XUAS-IVS-CsChrimson:tdTomato in VK00022; R71G01-Zp;GAL4DBD in attP2/+* |
| *Figure 5—figure supplement 2* | A | P1ª-GAL4, UAS > stop > CsChrimson (VK00022), fruᶠᴸᴾ | *w/Y; R15A01-p65AD:Zp in attP40/ 20XUAS > myr:TopHAT2 > CsChrimson: tdTomato in VK00022; R71G01-Zp; GAL4DBD in attP2/fruᶠᴸᴾ* |
| *Figure 5—figure supplement 2* | A | P1ª-GAL4, UAS > stop > CsChrimson (VK00005), fruᶠᴸᴾ | *w/Y; R15A01-p65AD:Zp in attP40/+; 71G01-Zp;GAL4DBD in attP2/ 20XUAS > myr:TopHAT2 > CsChrimson: tdTomato in VK00005, fruᶠᴸᴾ* |
| *Figure 5—figure supplement 2* | A | P1ª-GAL4, UAS > stop > CsChrimson (VK00005), dsxᶠᴸᴾ | *w/Y; R15A01-p65AD:Zp in attP40/+; 71G01-Zp;GAL4DBD in attP2/ 20XUAS > myr:TopHAT2 > CsChrimson: tdTomato in VK00005, dsxᶠᴸᴾ* |
| *Figure 5—figure supplement 2* | C, E | P1ª-GAL4 ∩ fruᶠᴸᴾ | *w/Y; R15A01-p65AD:Zp in attP40/+; 71G01-Zp;GAL4DBD in attP2/ 20XUAS > myr:TopHAT2 > CsChrimson: tdTomato in VK00005, fruᶠᴸᴾ* |
| *Figure 5—figure supplement 2* | D, E | NP2631 ∩ dsxᶠᴸᴾ | *w/Y; NP2631/ 20XUAS > myr:TopHAT2 > CsChrimson: tdTomato in VK00022; dsxᶠᴸᴾ/+* |
| *Figure 5—figure supplement 2* | B | P1ª-GAL4, UAS > stop > CsChrimson, fruᶠᴸᴾ | *w/Y; R15A01-p65AD:Zp in attP40/+; R71G01-Zp:GAL4DBD in attP2/ 20XUAS > myr:TopHAT2 > CsChrimson: tdTomato in VK00005, fruᶠᴸᴾ* |
| *Figure 5—figure supplement 2* | B | P1ª-GAL4, UAS > stop > CsChrimson | *w/Y; R15A01-p65AD:Zp in attP40/+; R71G01-Zp;GaL4DBD in attP2/ 20XUAS > myr:TopHAT2 > CsChrimson: tdTomato in VK00005* |
| *Figure 5—figure supplement 2* | B | UAS > stop > CsChrimson, fruᶠᴸᴾ | *w/Y; R15A01-p65AD:Zp in attP40/+; +/20XUAS > myr:TopHAT2 > CsChrimson: tdTomato in VK00005, fruᶠᴸᴾ* |
| *Figure 5—figure supplement 3* | A, B | NP2631 ∩ fruᶠᴸᴾ, R15A01-LexA | *w/Y; NP2631/13XLexAop2-IVS-GCaMP6f-p10 in su(Hw)attP5; R15A01-LexA in attP2/ 20XUAS > myr:TopHAT2 > CsChrimson: tdTomato in VK00005, fruᶠᴸᴾ* |
| *Figure 5—figure supplement 3* | C, D | NP2631 ∩ fruᶠᴸᴾ, R71G01-LexA | *w/Y; NP2631/13XLexAop2-IVS-GCaMP6f-p10 in su(Hw)attP5; 20XUAS > myr:TopHAT2 > CsChrimson: tdTomato in VK00005, fruᶠᴸᴾ/R71G01-LexA in attP2* |
| *Figure 6* *Figure 6—figure supplement 1* | A, E, F, H, I F | P1ª-GAL4, XY, *fru* locus: fruᴹ/fru⁴⁻⁴⁰ | *w/Y; R15A01-p65AD:Zp in attP40/ 20XUAS-IVS-CsChrimson:tdTomato in VK00022; R71G01-Zp:GAL4DBD in attP2, fru⁴⁻⁴⁰/fruᴹ* |
| *Figure 6* | B, E | P1ª-GAL4, XX, *fru* locus: fruᴹ/fru⁴⁻⁴⁰ | *w/w; R15A01-p65AD:Zp in attP40/20XUAS-IVS-CsChrimson:tdTomato in VK00022; R71G01-Zp:GAL4DBD in attP2, fru⁴⁻⁴⁰/fruᴹ* |

Table 1 continued on next page

*Table 1 continued*

| Figure | Panel | Abbreviated genotype | COMPLETE GENOTYPE ('Y' represents the Y chromosome) |
|---|---|---|---|
| *Figure 6* | C, E, G-I | P1ᵃ-GAL4, XY, *fru* locus: *fru^F^/fru^4-40^* | *w/Y; R15A01-p65AD:Zp in attP40/20XUAS-IVS-CsChrimson:tdTomato in VK00022; R71G01-Zp:GAL4DBD in attP2, fru^4-40^/fru^F^* |
| *Figure 6* | D, E | P1ᵃ-GAL4, XX, *fru* locus: *fru^F^/fru^4-40^* | *w/w; R15A01-p65AD:Zp in attP40/20XUAS-IVS-CsChrimson:tdTomato in VK00022; R71G01-Zp:GAL4DBD in attP2, fru^4-40^/fru^F^* |
| *Figure 6* | E, H, I | P1ᵃ-GAL4, XY, *fru* locus: +/+ | *w/Y; R15A01-p65AD:Zp in attP40/20XUAS-IVS-CsChrimson:tdTomato in VK00022; R71G01-Zp:GAL4DBD in attP2/+* |
| *Figure 6—figure supplement 1* | A, E | | |
| *Figure 6* | E | P1ᵃ-GAL4, XX, *fru* locus: +/+ | *w/w; R15A01-p65AD:Zp in attP40/20XUAS-IVS-CsChrimson:tdTomato in VK00022; R71G01-Zp:GAL4DBD in attP2/+* |
| *Figure 6—figure supplement 1* | B | | |
| *Figure 6* | E | P1ᵃ-GAL4, XY, *fru* locus: +/fru^4-40^ | *w/Y; R15A01-p65AD:Zp in attP40/20XUAS-IVS-CsChrimson:tdTomato in VK00022; R71G01-Zp:GAL4DBD in attP2, fru^4-40^/+* |
| *Figure 6—figure supplement 1* | C | | |
| *Figure 6* | E | P1ᵃ-GAL4, XX, *fru* locus: +/fru^4-40^ | *w/w; R15A01-p65AD:Zp in attP40/20XUAS-IVS-CsChrimson:tdTomato in VK00022; R71G01-Zp:GAL4DBD in attP2, fru^4-40^/+* |
| *Figure 6—figure supplement 1* | D | | |
| *Figure 6—figure supplement 1* | G | P1ᵃ-GAL4, UAS-CsChrimson, *fru* locus: +/+ | *w/Y; R15A01-p65AD:Zp in attP40/20XUAS-IVS-CsChrimson:tdTomato in VK00022; R71G01-Zp:GAL4DBD in attP2, +/+* |
| *Figure 6—figure supplement 1* | G | P1ᵃ-GAL4, UAS-CsChrimson, *fru* locus: +/fru^4-40^ | *w/Y; R15A01-p65AD:Zp in attP40/20XUAS-IVS-CsChrimson:tdTomato in VK00022; R71G01-Zp:GAL4DBD in attP2, fru^4-40^/+* |
| *Figure 6—figure supplement 1* | G | P1ᵃ-GAL4, UAS-CsChrimson, *fru* locus: *fru^M^/fru^4-40^* | *w/Y; R15A01-p65AD:Zp in attP40/20XUAS-IVS-CsChrimson:tdTomato in VK00022; R71G01-Zp:GAL4DBD in attP2, fru^4-40^/fru^F^* |
| *Figure 6—figure supplement 2* | A-C | fruF ♂, P1ᵃ-GAL4, UAS-CsChrimson | *w/Y; R15A01-p65AD:Zp in attP40/20XUAS-IVS-CsChrimson:tdTomato in VK00022; R71G01-Zp:GAL4DBD in attP2, fru^4-40^/fru^F^* |
| *Figure 6—figure supplement 2* | A | fruF ♂, P1ᵃ-GAL4 | *w/Y; R15A01-p65AD:Zp in attP40/10XUAS-IVS-GFP-p10 in VK00022; R71G01-Zp:GAL4DBD in attP2, fru^4-40^/fru^F^* |
| *Figure 6—figure supplement 2* | A | fruF ♂, UAS-CsChrimson | *w/Y; R15A01-p65AD:Zp in attP40/20XUAS-IVS-CsChrimson:tdTomato in VK00022; fru^4-40^/fru^F^* |
| *Figure 6—figure supplement 2* | C | ♂, P1ᵃ-GAL4, UAS-CsChrimson | |

sev^21^; P{CaryP}attP2 (RRID:BDSC_25710) strain (BestGene Inc, Chino Hills, CA), and transformants were recovered using on the mini-*white* eye color marker.

## Immunohistochemistry

The following antibodies were used for immunohistochemistry with dilution ratios as indicated: rabbit anti-DsRed (1:1,000, Clontech # 632496, RRID:AB_10013483), mouse anti-BRP (1:100; Developmental Studies Hybridoma Bank nc82 (concentrated), RRID:AB_2314866), chicken anti-GFP (1:1,000, Abcam # ab13970, RRID:AB_300798), rabbit anti-FruM (1:10,000, a gift from Barry Dickson; *Stockinger et al., 2005*), guinea pig anti-FruM (1:100), rat anti-DsxM (1:100) (both gifts from Michael Perry (University of California, San Diego)), goat anti-chicken Alexa 488 (1:100, ThermoFisher Scientific Cat# A11039, RRID:AB_2534096), goat anti-rat Alexa 488 (1:100, ThermoFisher Scientific Cat# A11006, RRID:AB_2534074), goat anti-rabbit Alexa 568 (1:100; ThermoFisher Scientific Cat# A11036, RRID:AB_10563566), goat anti-mouse Alexa 633 (1:100; ThermoFisher Scientific Cat# A21052, RRID:AB_2535719), goat anti-guinea pig Alexa 633 (1:100, ThermoFisher Scientific Cat# A21105, RRID:AB_2535757). Immunohistochemistry of the fly brains followed the protocol described in *Van Vactor et al. (1991)*. Briefly, the fly brains are dissected in 1XPBS and fixed in 1XPBS with 2%

formaldehyde and 75 mM L-lysine for 75–90 min at room temperature. The brains were then washed in PBST (1XPBS, 0.3% TritonX-100) and were incubated in the blocking solution (10% heat-inactivated normal goad serum, 1XPBS, 0.3% TritonX-100) for 30 min. Primary antibodies were diluted in the blocking solution and were applied to samples, which were then incubated at 4°C for 2 days. The brains were then washed in PBST and then incubated in the blocking solution for 30 min. Secondary antibodies were diluted in the blocking solution and were applied to the samples, which were then incubated at 4°C overnight. The brains were then washed in PBST, and then either incubated in 1XPBS, 50% glycerol for 2 hr at room temperature before mounted in Vectashield (Vector Laboratories, Cat# H-1000) onto a slide glass, or incubated in FocusClear (CelExplorer Labs, Taiwan, Cat# FC-101) medium for 2 hr at room temperature before being mounted in MountClear (CelExplorer Labs, Taiwan, Cat# MC-301) medium. A small well was made by cutting vinyl tape fixed on a slide glass, and one brain was transferred to each well before a cover slip (#1.5) was placed on the well and was sealed with nail polish. All reactions were carried out in a well of 6 × 10 microwell mini tray (ThermoFisher Scientific Cat# 439225).

For simultaneous detection of DsxM and FruM (*Figure 2—figure supplement 2B,C*, *Figure 5—figure supplement 1C,D*), brains were fixed in 1XPBS with 4% formaldehyde for 15 min at room temperature. Brains were then transferred to a 1.5 mL microtube, in which the remaining steps were carried out.

All z-stack images were acquired by FV-1000 confocal microscopy (Olympus America; kindly shared by Dr. Samuel Pfaff at Salk Institute) except samples for *Figure 5C,D* and *Figure 5—figure supplement 3C,D*, which were acquired by a Zeiss 710 confocal microscopy (Carl Zeiss Microscopy) at the Salk Institute Biophotonics Core, and were processed in Fiji software (*Schindelin et al., 2012*) (RRID:SCR_002285; https://fiji.sc/). The despeckle function was applied before a z-projection image was generated using maximum intensity projection. Minimum or maximum intensity thresholds were adjusted for enhanced clarity. Source image files used in all figures can be found in *Source data 1*.

## Registration and analysis of immunohistochemical samples

All data points for anatomical quantifications used in all figures, as well as all statistical results with exact p values, can be found in *Source data 1*.

Parametric tests were applied as indicated in figure legends to compare cell body numbers among different genotypes. All data points have biological replicates of at least 8, which has sufficient power to detect mean changes in cell body number of larger than 1.75 when assuming a mean cell body number of 5 and a standard deviation of 1.5 (which is reasonable based on our data). As indicated in figures, genotype-dependent changes in cell body number had a larger effect size than two in all cases.

We first split each channel of a z-stack file and resaved each as individual. nrrd files in Fiji. We then used the Fiji plugin for Computational Morphometry Tookit (CMTK) (RRID:SCR_002234; https://www.nitrc.org/projects/cmtk) (*Rohlfing and Maurer, 2003*) to register the brain image stacks to the template brain as described in *Jefferis et al. (2007)* (https://github.com/jefferis/fiji-cmtk-gui). Briefly, for each brain, the image of neuropil visualized by anti-BRP antibody was used to transform the z-stack to the template brain, and the same transformation was subsequently applied to additional channels. Registration was performed by using the same parameters implemented in '*Cachero et al. (2010)*' (exploration = 26, coarsest = 8, grid spacing = 80, refine = 4, accuracy = 0.4) in CMTK plugin on Fiji. We used JRC2018 INTERSEX (*Bogovic et al., 2018*) as our template, since we experienced more robust registration result than with the FCWB template brain (*Costa et al., 2016*) (data not shown).

After the registration of z-stacks, we calculated the average of the images in a hemibrain for each genotype. We did this by horizontally flipping each transformed z-stack image, and calculated the average signal intensity in each voxel. The resulting averaged images are bilaterally symmetrical. Therefore, only the left hemisphere is shown for z-projection images.

To calculate the volumes of specific neuronal structures, we first visualized each z-stack in a 3D space using a rendering software FluoRender (*Wan et al., 2009*) (RRID:SCR_014303; https://github.com/SCIInstitute/fluorender). We then segmented the target structures using the Paint Brush function and calculated the volume of each structure using the Volume Size function (threshold = 800) (*Wan et al., 2017*). Statistical analyses were carried out using MATLAB (The Mathworks, Inc, RRID:SCR_001622). The Kruskal-Wallis test ('kruskalwallis') was used to evaluate whether a volume of the

given structure was significantly different among different genotypes. When the p-value was below 0.05, the post-hoc the Mann-Whitney U-test ('ranksum') was used to detect significant differences between testing and control genotypes. In both cases, the Bonferroni correction was applied to p values. Non-parametric tests were applied for volume data since we could not necessarily assume a normal distribution for this data type.

## Social behavior analysis

### Subject preparation

Flies were collected on the day of eclosion into vials containing standard cornmeal-based food, and were kept either as a group of up to 16 flies per vial, or singly at 25 °C with 60% relative humidity, and a 9AM:9PM light:dark cycle. For optogenetic experiments, the tester flies were reared on food containing 0.2 mM all-*trans* retinal (MilliporeSigma, Cat#R2500, 20 mM stock solution prepared in 95% ethanol), and vials were covered with aluminum foil to shield light. Every 3 days, flies were transferred to vials containing fresh food. Tester flies for *Figures 1–4* (including figure supplements) and *Figure 5—figure supplements 2* and *3* were aged for 14–16 days to ensure consistent labeling of targeted neurons (data not shown). Tester flies for *Figures 5* and *6*, and *Figure 5—figure supplement 1* were aged for 5–7 days.

Male target flies were group-reared Canton-S (originally from the lab of Martin Heisenberg, University of Würzberg) virgin males. To prepare mated wild-type target females, 5 Canton-S males were introduced into vials with 10 virgin females at 4 days old, and were reared for two more days to let them mate. At 3 days old, both male and female target flies were briefly anesthetized with $CO_2$, and the tip of either one of their wings were clipped by a razor to create a 'mark'. This clipping treatment did not reduce the amount of each behavior (lunge, wing extension, and headbutt) detected under our experimental settings (*Figure 1—figure supplement 2B*).

### Behavioral assays

All behavior assays were conducted in the evening (from 4 to 9PM) at 22–25°C. Social behavior assays were performed in a '12-well' acrylic chamber (*Asahina et al., 2014*) with food substrate (apple juice (Minute Maid) supplemented with 2.25% w/v agarose and 2.5% w/v sucrose; *Hoyer et al., 2008*) covering the entire floor of arena. The wall was coated with Insec-a-Slip (Bioquip Products, Inc, Cat# 2871C) and the ceiling was coated with Surfasil Siliconizing Fluid (ThermoFisher Scientific, Cat# TS-42800), both to prevent flies from climbing, as described previously (*Asahina et al., 2014*; *Hoyer et al., 2008*). The arenas were lit by LED backlights, which were controlled by a custom-built switch box. For optogenetic experiments, 850 nm infrared LED backlights (Sobel Imaging Systems, CA, Cat# SOBL-150 × 100–850) were used. Flies were introduced into the chamber by gentle aspiration, and were allowed to acclimate for 5 min before recording started.

Recording was done by USB3 digital cameras (Point Grey Flea3 USB3.0, FLIR Inc, Cat# FL3-U3-13Y3M-C) controlled by the BIAS acquisition software (IORodeo, CA; https://bitbucket.org/iorodeo/bias). The camera was equipped with a machine vision lens (Fujinon, Cat# HF35HA1B), and an infrared longpass filter (Midwest Optical Systems, Cat# LP780-25.5) when the infrared light sources were used. Movies were taken at 60 frames per second in the AVI format, either for 10 min in the optogenetic experiments or for 30 min for non-optogenetic experiments. Flies were discarded after each experiment. The food substrates were changed to a new one after 2 recordings for 30 min movies, or after 3 recordings for 10 min movies.

The setup for optogenetic experiments was assembled as described previously (*Inagaki et al., 2014*). Briefly, the red light (655 nm) LEDs were controlled via an Arduino Uno board (Arduino, Italy) using a custom program. As illustrated in *Figure 1D*, the stimulation paradigm (10 min in total) consists of 1 min pre-stimulation (time window '1' in *Figure 1D*), three blocks of 1 min stimulation at an indicated frequency (time window '2', 3 min in total) each followed by 2 min inter-stimulus intervals (ISIs, time window '3', 6 min in total). The recording and LED control were manually started simultaneously.

### Creation of behavioral classifiers

Lunges, wing extensions, and headbutts were quantified using behavioral classifiers developed in JAABA (*Kabra et al., 2013*) (https://sourceforge.net/projects/jaaba/files/), which runs on MATLAB.

The details of the classifier development and quantification of performance will be described elsewhere (Leng X, Wohl M, Ishii K, Nayak P, and Asahina K, manuscript in preparation). Briefly, we trained the lunge classifiers using 20 male pair movies, nine male-female pair movies, five female pair movies, five fruM female-fruF male pair movies, and five male-fruF male pair movies; wing extension classifiers using 28 male pair movies, 12 male-female pair movies, eight female pair movies, and three fruM female-fruF male pair movies; headbutt classifier using 15 male pairs movies, seven male-female pair movies, 13 female pair movies, four fruM female-fruF male pair movies, and two male-fruF male pair movies. We then evaluated the performance of these classifiers using 17 male pair movies (223.5 min in total), four male-female pair movies (120 min in total), five female pair movies (150 min in total), and four fruM female-fruF male pair movies (34 min). These pairs were not used as training samples, and their behaviors were manually annotated prior to evaluation independently by at least two people (referred to as 'true' behaviors).

Recall rate was calculated as: (number of 'true' behavior bouts detected by a classifier)/(total number of 'true' behavior bouts).

While precision rate was calculated as: (number of 'true' behavior bouts detected by a classifier)/[(total number of behavior bouts detected by a classifier) + (total number of false positive bouts detected by a classifier)].

## Quantification of social behavior data

All behavioral data points used in all figures, as well as all statistical results with exact p values, can be found in *Source data 1*.

Acquired movies were first processed by the FlyTracker program (*Eyjolfsdottir et al., 2014*) (http://www.vision.caltech.edu/Tools/FlyTracker/), which runs on MATLAB (The Mathworks, Inc). The regions of interest were defined as circles corresponding to the chamber of each arena. The identities of tester and target flies were manually confirmed. The fly pair was removed from further analysis when (1) one of the two flies were killed during introduction to the chamber, (2) the wings of either fly was stuck at the extended position, or (3) the discrimination of the two flies was impossible due to wing damage of a tester fly. The amount of behavior is the number of bouts for lunges and headbutts, and the total duration (seconds) for wing extensions. These amounts were binned per minute for quantification. Extremely short bouts detected by a classifier were almost always false positives, and were eliminated from quantification. For lunges and headbutts, events with duration of less than 50 milliseconds were discarded. For wing extensions, events with duration of less than 100 milliseconds were discarded. The post-processing of data was done using custom MATLAB codes which are available on Github (https://github.com/wohlmp/Ishii_Wohl_DeSouza_Asahina_2019; *Wohl, 2019*; copy archived at https://github.com/elifesciences-publications/Ishii_Wohl_DeSouza_Asahina_2019).

The duration of time a tester fly orients toward a target fly ('time orienting') was defined as the duration in which the following three conditions are met: (1) a target fly is within ±60° of the heading direction of a tester fly, (2) the distance between the ellipsoid centers of a tester and a target fly is within 5 mm (approximately two fly body length), and (3) the moving speed of the tester fly was above 0.1 mm/sec. The second and third conditions were used to eliminate frames in which a tester fly was standing still (and often grooming) while the target fly walked past in front of the tester fly. Frames that simultaneously fulfilled these three conditions were directly calculated from the output file of the tracker ('trx.mat' file) using MATLAB. To further eliminate incidents caused by erroneous flipping of fly orientations and other tracking errors, bouts with a duration of less than 200 milliseconds were discarded. This time period includes 'chasing', which has been observed in the context of both courtship and aggressive behaviors (*Chen et al., 2002*; *Dankert et al., 2009*; *Hall, 1994*).

The frame in which the infrared indicator LED turned on during the first LED stimulation period was used to align frames of movies. Statistical analyses were carried out using MATLAB. After behavior within each time window were calculated (see *Figure 1D*), the Kruskal-Wallis test ('kruskalwallis') was used to evaluate whether a given behavior was significantly different among different illumination windows (windows 1, 2, and 3; *Figure 1D*) or among different genotypes. When the p-value was below 0.05, the post-hoc Mann-Whitney signed rank test ('signrank') was used to detect significant differences between illumination period, and the Mann-Whitney U-test ('ranksum') was used to detect significant differences between testing and control genotypes. In both cases, Bonferroni

correction was applied to p values. When the uncorrected p value was less than 0.05 but the corrected value did not pass the significant level, the uncorrected value was shown on panels with parenthesis.

In *Figure 2—figure supplement 2D2*, each tester fly was classified as 'lunge increased' in window two relative to window 1 or three if (1) lunges per minute increased more than 50%, or (2) more than one lunge per minute was performed when no lunge was detected in either window 1 or 3. In *Figure 2—figure supplement 2D3*, each tester fly was classified as 'wing extension increased' in window two relative to window 1 or three if (1) wing extension duration per minute was increased more than 50%, or (2) more than 500 milliseconds per minute of wing extension was performed when no wing extension was detected in either window 1 or 3. Fisher's exact test ('fishertest') was used to determine whether the frequency of flies that increased lunging or wing extension was significantly different from the frequency of brains with a descending neuron (as shown in $D_1$).

Generally, noticeably less lunges were detected in fruF males compared to in males. Often the corresponding classifiers did not detect any behaviors. In such zero-inflated datasets, a few false positive incidents can greatly impact the result of statistical tests. To avoid this pitfall, classifier results of lunges from fruF males were manually validated, and obvious false positives (caused by tracking errors, when a fly was near or on the wall, or when a fly suddenly jumped) were eliminated before statistical tests were applied.

## Acknowledgements

We thank Drs. David Anderson, Gerald Rubin, and Barret Pfeiffer for sharing unpublished transgenic *Drosophila* strains with us; Drs. Stephen Goodwin and Daisuke Yamamoto for other *Drosophila* strains; Eyrun Eyjolfsdottir and Dr. Pietro Perona for developing and improving the FlyTracker program; Dr. Michael Perry for sharing with us anti-DsxM and anti-FruM antibodies; Dr. Yong Wan for development and support for FluoRender, Pavan Nayak and Vivian Shaw for their help on the development of behavior classifiers, Dr. Samuel Pfaff for sharing the Olympus FV-1000 confocal microscopy with us, Drs. Eiman Azim, Weizhe Hong, Samuel Pfaff, Carla Shatz, John Thomas, and members of the Asahina lab for critical comments on the manuscript, and David O'Keefe for scientific editing on the manuscript. The antisera nc82 (anti-BRP), developed by E Buchner, were obtained from the Developmental Studies Hybridoma Bank, created by the NICHD of the NIH and maintained at The University of Iowa, Department of Biology, Iowa City, IA 52242. Stocks obtained from the Bloomington *Drosophila* Stock Center (NIH P40OD018537) were used in this study. This work was also made possible in part by software funded by the NIH: FluoRender: Visualization-Based and Interactive Analysis for Multichannel Microscopy Data, 1R01EB023947-01 and the National Institute of General Medical Sciences of the National Institutes of Health under grant number P41 GM103545-18.

## Additional information

### Funding

| Funder | Grant reference number | Author |
| --- | --- | --- |
| National Institute of General Medical Sciences | GM119844 | Kenichi Ishii Margot Wohl Kenta Asahina |
| Naito Foundation | | Kenichi Ishii |
| Japan Society for the Promotion of Science | | Kenichi Ishii |
| Mary K. Chapman Foundation | | Margot Wohl |
| Rose Hills Foundation | | Margot Wohl |

The funders had no role in study design, data collection and interpretation, or the decision to submit the work for publication.

## Author contributions
Kenichi Ishii, Andre DeSouza, Investigation; Margot Wohl, Software, Investigation; Kenta Asahina, Conceptualization, Resources, Data curation, Formal analysis, Supervision, Funding acquisition, Validation, Investigation, Visualization, Methodology, Project administration

## Author ORCIDs
Kenta Asahina (iD) https://orcid.org/0000-0001-6359-4369

## Decision letter and Author response
Decision letter https://doi.org/10.7554/eLife.52701.sa1
Author response https://doi.org/10.7554/eLife.52701.sa2

# Additional files

## Supplementary files
• Source data 1. This file contains data points represented in all figure panels, as well as p-values for all statistical tests shown in the figures.

• Supplementary file 1. Key Resources Table.

• Transparent reporting form

## Data availability
All data and statistical results are available in an accompanying source data file.

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
