## [Decision Letter]

**Acceptance summary:**

The evidence in this paper broadly supports the interesting conclusion that two distinct classes of sexually dimorphic neurons control male-specific courtship behaviour and its targeted display towards females. This is concluded based on the different requirement of *Doublesex* and *Fruitless* for each of these aspects of male courtship and the evidence that they function in different subsets of neurons.

**Decision letter after peer review:**

[Editors’ note: the authors submitted for reconsideration following the decision after peer review. What follows is the decision letter after the first round of review.]

Thank you for submitting your work entitled "A genetic code that specifies sexually dimorphic circuits and their control of social behaviors" for consideration by *eLife*. Your article has been reviewed by a Senior Editor, a Reviewing Editor, and three reviewers. The reviewers have opted to remain anonymous.

Our decision has been reached after consultation between the reviewers. In the discussions, all the referees expressed interest in the main message (dissociating *fru* and *dsx* functions and relating these to vertebrate mechanisms) as well as enthusiasm for several very interesting and clear findings from an impressively extensive set of experiments. However, there were also substantial concerns leading to the consensus opinion that additions and revisions required were too extensive to be accomplished in the two months required by *eLife*.

Based on these discussions and the individual reviews appended below, we regret to inform you that, in present form, your work will not be considered further for publication in *eLife*. However, if you choose to revise the paper on the lines suggested by the referees and resubmit, then *eLife* would be happy to consider a fresh submission (or two) that we would endeavour to have reviewed by the same referees.

In addition to the specific comments below, the referees offered some specific overall suggestions for revision that you may find useful. These include: (a) to refocus the paper on solely on the *fru/dsx* story; (b) to include new rigorous anatomical analysis of the "P1" cluster in P1^a^ and NP2631 flies as requested in the reviews; and (c) providing a detailed discussion of limitation of using systemic/constitutive alleles of *fru/dsx* in relation to behavior as well as anatomy. The referees also felt that the story be clarified and better communicated by splitting into two separate but simpler manuscripts, with a potentially a separate manuscript on Tk, but you should of course make your own decision on this rather too interventional opinion.

Reviewer #1:

In both flies and mammals, multiple sexual differentiation mechanisms guide display of adult social behaviors. In flies it is *dsx* and *fru*, whereas in mammals it is largely testosterone and estrogen. Ishii et al. present a wide-ranging set of observations that they use to suggest that various aspects of sexually dimorphic courtship and aggression displays in flies are controlled by either *dsx* or *fru*. This seems an over-interpretation of the findings. I am supportive of the study but have a few suggestions.

1) Data from Figure 3 is used to conclude that *dsx* specifies both P1 and NP2631 neurons. This is inconsistent with the anatomy shown in figure panels, which shows that both *dsx* and *fru* contribute. It is true that cell number is dependent on *dsx*, but the morphology of the arbors indicates that *fru* is essential as well: E3/J3/K3 are different than C3/H3/I3. Similarly, data from Figure 6 is used to conclude that fruM but not *dsx* specifies Tk neurons; comparison of E3 and F3 indicates that dsxM is also important. The authors should quantify arborization rather than rely on gross examination for Figure 3 and Figure 6.

Presentation of the histology could be better elsewhere too. For example, from panels Figure 2A,C, I conclude that there is no overlap of *R15A01* and *R71G01*, raising the question of the intersectional identification of P1^a^ neurons.

2) It is clear that FruM is essential for male type lunges toward other males and enhancing wing extension to females. However, FruM is also essential for the pattern of wing extension upon activation of P1^a^ neurons; in controls (Figure 1), wing extensions in phase 2>3 and 2<3 toward males and females, whereas in FruF males (Figure 4A-D), wing extensions in phase 2<3 toward males and females. In addition, it is clear that FruM is essential for regulating intensity of wing extensions. It is a gross simplification to conclude that dsxM enables execution of male courtship whereas FruM enforces choice of target sex. Clearly, FruM is also critical for execution of this behavior.

3) Why do males not lunge toward other males in the absence of activation of P1^a^ or NP2631 or Tk neurons (Figure 1, Figure 6)?

4) There is some over-simplification of the literature. It might be true in flies that most studies have only used one sex as a target individual to analyze function of particular neurons (subsection “Effect of a target fly’s sex on optogenetically induced social interactions”), but this is certainly not true in mice (studies from Anderson, Dulac, Lin, Shah).

Rather than draw parallels to the four core genotype in the mouse (using which not a single gene has been identified that regulates social behavior independent of sex hormones, subsection “*dsx*, and not *fru*, specifies both P1^a^ and NP2631 ∩ dsx^FLP^ neurons”), it would be more appropriate to discuss the role of estrogen vs. testosterone in regulating male behaviors. It is clear that estrogen masculinizes the brain during a critical window whereas testosterone signaling amplifies male behaviors in adult life. This would be a more instructive analogy when trying to dissect the specific functions of *dsx* and *fru*.

Reviewer #2:

Unfortunately, I have some major technical concerns with the manuscript, and the lack of quantitative anatomical analyses throughout. The rather cursory analysis offered in this manuscript has several critical caveats that are not discussed and undermine the conclusions that can be drawn from the experimental data. Therefore, this manuscript is not suitable for publication without significant experimental clarification of these points.

Subsection “Effect of a target fly’s sex on optogenetically induced social interactions” and throughout the manuscript:

Neither the P1^a^ split-Gal4 nor NP2631 ∩ dsx^FLP^ driver are 'P1/pC1'-specific, some additional neurons in the brain express Gal4 in both cases (see both Hoopfer et al., 2015 and Koganezawa et al., 2016). Therefore, by using these drivers, the authors cannot conclude that the observed behavioral phenotypes were caused purely by the purported 'P1/pC1' neurons. Further intersectional approaches to rigorously target these neurons are essential.

Subsection “*dsx*, and not *fru*, specifies both P1^a^ and NP2631 ∩ dsx^FLP^ neurons”:

The expression 'P1^a^ neurons were specified only in the flies that had dsxM' is not accurate and is likely even wrong. Whether the presence of the P1 neurons depends on *fru* or *dsx* was originally examined in the very first P1 paper, Kimura et al., 2008. Kimura et al. previously demonstrated by using *dsx* mutations that the absence of dsxF, but not the presence of dsxM, is the most striking factor for determining the presence of the P1 neurons.

In Figure 3, the *dsx* locus was always wild type and therefore the authors could not dissociate the presence of dsxM and the absence of dsxF. It is essential for the authors to do the similar experiments in combination with the appropriate *dsx* mutant alleles.

The authors mentioned that 'processes of P1^a^ neurons in fruF males appeared thinner than those in males'. What does thinner mean? What form of scientific measurement is "thinner"? This needs to be quantified properly. Image co-registration or any other quantitative analyses for the size differences of the neuronal processes between these genotypes must be carried out.

The authors also mention ‘sex differences both in terms of cell body numbers and branching patterns’. If the authors declare the branching patterns were different between genotypes, they must carry out quantitative analyses.

They also declare the neuronal morphology was 'similar' between particular genotypes, but what does 'similar' mean? The authors seem to conclude that the neuronal morphology is different when they expect it is different and that it is similar when they expect it is similar, without any quantitative criteria for measurement.

Subsection “P1^a^ and NP2631 ∩ dsx^FLP^ neurons in fruF males can induce courtship, but not male-type aggressive behaviour”:

The logical link between the anatomy in Figure 3 and behavior in Figure 4 is unclear. In Figure 3, the authors looked at the contributions of the *fru* and *dsx* genes in the purported 'P1/pC1'-specific neurons, but in Figure 4, they carried out behavioral assays with the *fru* mutants, in which *fru* gene function is disrupted not only in P1/pC1 neurons but also all other FruM-expressing neurons throughout the PNS and CNS, they then base all their conclusions on only the behavioral function of P1/pC1 neurons. These animals will have a plethora of *fru* mutant phenotypes that are associated with lack of FruM in other neurons in the CNS and PNS.

To make any relevant conclusions the authors need to re-do the behavioral assays in Figure 4 using the flies in which only the P1/pC1 neurons are mutant for *fru* (e.g., mutant MARCM clones, UAS-Cas9 system, or fru-RNAi).

Subsection “fruM specifies aggression-promoting Tk-GAL4^FruM^ neurons”:

The logic of transition from P1/pC1 to Tk-Gal4 neurons is very difficult to follow. The relationship between the former and the latter need to be explained. Are they completely different stories?

Figure 6:

As a control, the authors should demonstrate that Tk-GAL4 neurons do not express *dsx*.

Subsection “Differential roles of *fru* isoforms on male-male interactions and specification of Tk-GAL4^FruM^ neurons”:

The authors showed that the Tk-Gal4 neurons exist in fru^∆B^ mutant males but behavior is affected in these flies. The simplest explanation would be that the neurites are malformed in these flies.

Although the authors mentioned that 'fruMA and fruM mutants did not affect […] their arborization pattern', they didn't seem to carry out any quantitative morphological analyses. Such analyses should be included.

Subsection “The Tk-GAL4^FruM^ neurons in fruMB mutants can induce male aggression”:

The authors mention that 'these mutants have Tk-GAL4^FruM^ neurons that appear to retain their morphology'. But how did they examine this?

Reviewer #3:

This paper is a paradox. As far as I can tell it is rigorous on all levels. The scholarship is excellent, deep and well rounded. In addition, the issues are explored thoroughly and there is an enormous amount of data. In addition, an enormous number of fly lines (the table that describes the lines is phenomenal). But on the other hand, it is too much. I would like to discuss the argument and conclusions that surface relating *dsx* and *fru* expressing neurons with sexually dimorphic choice vs. sexually dimorphic motor patterns and the interesting differences between reproductive behavior and aggression, *but* the paper is simply too dense at almost every point. Reading it is a painful experience and this does the work a tremendous disservice. I say this recognizing and admiring the care that has gone into preparing this text. It seems flawless, just impossibly dense.

What to do? Here is a list of suggestions.

1) The text and figures are impeccably constructed but for the average reader they are overly complicated and far too dense. The flow of the manuscript would benefit greatly from severe editing of the text, figures and figure legends. It is with a heavy heart that I ask the authors to do this knowing the amount of effort it took to construct the manuscript.

2) Remove raster plots from figures throughout and move to supplemental material if needed. See Figure 1, Figure 4, Figure 5, Figure 6, Figure 7, Figure 9.

3) Figure 2 is not required. Move to supplemental material and concisely state result in text.

4) Much of the immunohistochemistry could be removed from figures and moved to supplemental material. See Figure 3, Figure 6, Figure 8.

5) The summary panels at the end of some of the figures are helpful. But a single unifying summary/model would be best. Though organized in a systematic and rational way the results still feel piecemeal and it is tiresome to continually have refer back to previous figures or supplemental material to regain your bearing. A single model would help immensely.

Alternatively, it might make sense to turn this into two papers, one about sex, the other about aggression?

[Editors’ note: further revisions were suggested prior to acceptance, as described below.]

Thank you for submitting your article "Sex-determining genes distinctly regulate courtship capability and target preference via sexually dimorphic neurons" for consideration by *eLife*. Your article has been reviewed by K VijayRaghavan as the Senior Editor, a Reviewing Editor,, and two reviewers. The reviewers have opted to remain anonymous.

The reviewers have discussed the reviews with one another and the Reviewing Editor has drafted this decision to help you prepare a revised submission.

Summary:

The evidence in this paper broadly supports the interesting conclusion that two distinct classes of sexually dimorphic neurons control male-specific courtship behaviour and its targeted display towards females. This is concluded based on the different requirement of *Doublesex* and *Fruitless* for each of these aspects of male courtship and the evidence that they function in different subsets of neurons.

Essential revisions:

Despite the notable revisions, e.g. valuable inclusion of quantitative anatomical analyses of the P1/pC1 neurons, there are remain issues that remain to be addressed.

1) Although the cover letter refers to Figure 1E of Hoopfer et al. (2015) to justify the view that the driver is P1/pC1-specific, Figure 1—figure supplement 1C of Hoopfer et al. (2015) appears to show that there are several additional neurons close to the lateral horn labelled by this split-Gal4 driver in both males and females. The main text of Hoopfer et al. (2015) states that "in addition to P1 neurons, the P1^a^ spGAL4 driver labelled a minor and variable population of neurons (<4 per hemi- brain) with cell bodies in the lateral horn (LH) that were neither FruM-positive nor male-specific".

2) The same issue applies to the NP2631 ∩ dsx^FLP^ driver. The text and cover letter reference Figure 4D of Koganezawa et al. (2016) to justify that their driver is P1/pC1-specific. However, the referenced figure appears to show a single descending axon per hemisphere is going out of the ventral end of the GNG. This axon crosses the midline at the neuropil region just dorsal to the oesophageal foramen, which is a typical neurite structure of another *dsx*-expressing neuronal cluster called pMN1 (see Figure S1 of Kimura et al., 2015). Similarly, in Figure S3A of Koganezawa et al. (2016), a descending axon of pMN1 is more obvious in this female specimen (pMN1 is present in both sexes with dimorphic neurite morphology, according to Figure S1 of Kimura et al., 2015). I would encourage the authors to show the expression patterns in the VNC. Are there any cells in this tissue labelled by the NP2631 ∩ dsx^FLP^ driver?

The above two Points 1 and 2 reflect concerns on the specificity of driver lines used. To address ambiguities that arise from this, unless there are additional strong arguments in support of the authors' interpretations, ideally all behavioural assays should be re-done under the conditions of P1/pC1-specific activation using either further intersection or a stochastic/mosaic approach.

3) It still remains unclear how *fru* function is dissociated from that of *dsx* in the sexual differentiation of P1/pC1 neurons. To clarify potentially distinct contributions of *fru* and *dsx* to this developmental process, it appears that *dsx* mutant analyses are required. Have these been attempted or done?

4) The experiments use fruM, fruF and single-isoform *fru* mutant alleles for behavior and attribute the behavioural phenotypes to P1/pC1 (this paper) and Tk-GAL4^FruM^ neurons (the other paper, titled “Layered roles of *fruitless* isoforms in specification and function of male aggression-promoting neurons in *Drosophila*”). Given that Fru isoforms are expressed in other neurons in the CNS and the PNS, surely the effects of Fru isoform on behavior in P1/pC1 and Tk-GAL4^FruM^ neurons can only really be assessed by looking at animals were the specific neurons are mutant for *fru* (e.g. *fru* isoform-specific mutant MARCM clones, UAS-Cas9 system, or fru-RNAi)? Otherwise, the data would need to be interpreted with obvious caveats.

5) The paper should include better images of *Dsx* immunolabeling (Figure 2—figure supplement 2E) as the current one is of very poor quality.

---

## [Author Response]

[Editors’ note: the authors resubmitted a revised version of the paper for consideration. What follows is the authors’ response to the first round of review.]

Our decision has been reached after consultation between the reviewers. In the discussions, all the referees expressed interest in the main message (dissociating fru and dsx functions and relating these to vertebrate mechanisms) as well as enthusiasm for several very interesting and clear findings from a impressively extensive set of experiments. However, there were also substantial concerns leading to the consensus opinion that additions and revisions required were too extensive to be accomplished in the two months required by eLife.Based on these discussions and the individual reviews appended below, we regret to inform you that, in present form, your work will not be considered further for publication in eLife. However, if you choose to revise the paper on the lines suggested by the referees and resubmit, then eLife would be happy to consider a fresh submission (or two) that we would endeavour to have reviewed by the same referees.In addition to the specific comments below, the referees offered some specific overall suggestions for revision that you may find useful. These include: (a) 1 to refocus the paper on solely on the fru/dsx story; (b) to include new rigorous anatomical analysis of the "P1" cluster in P1^a^ and Np2631 flies as requested in the reviews; and (c) providing a detailed discussion of limitation of using systemic/constitutive alleles of fru/dsx in relation to behavior as well as anatomy. The referees also felt that the story be clarified and better communicated by splitting into two separate but simpler manuscripts, with a potentially a separate manuscript on Tk, but you should of course make your own decision on this rather too interventional opinion.

We decided to split our original manuscript into two, following two reviewers’ suggestions that this would increase clarity. We agree that our attempts to integrate our findings on 2 genes (*dsx* and *fru*), 2 behaviors (courtship and aggression), and 3 neuronal populations (NP2631 ∩ dsx^FLP^, P1^a^, and Tk-GAL4^FruM^ neurons) caused difficulties with the flow of logic. We believe that two separate manuscripts allow us to focus on two relatively simpler messages individually, and help us present relevant data without causing confusion. The current paper, “Sex-determining genes distinctly regulate courtship capability and target preference via sexually dimorphic neurons”, also serves to incorporate your first major recommendation: “to refocus the paper on solely on the *fru/dsx* story”. In line with this recommendation, we now include a new set of data showing that *dsx* and *fru*-co-expressing neurons in males can promote both courtship and aggressive behaviors in a target sex-dependent manner (see Figure 1 of this manuscript).

Regarding your second recommendation to provide rigorous neuroanatomical analysis, we thoroughly re-evaluated our anatomical data by registering brains to a common template and analyzing specific volumes with 3-dimentional segmentation. Please see Figure 3L-M, Figure 6H, I, in “Sex-determining genes distinctly regulate courtship capability and target preference via sexually dimorphic neurons”, and Figure 1N-P, and Figure 5I-K in “Layered roles of *fruitless* isoforms in specification and function of male aggression-promoting neurons in *Drosophila*” for new quantification of neuronal morphologies. Standardized immunohistochemical data enabled us to directly compare the morphology of specific neurons across multiple genotypes. Moreover, we could detect statistical differences in neuroanatomy by quantitatively comparing segmented volumes of specific neuronal structures. This rigorous quantification revealed differences in P1^a^ neurons between males and fruF males, which compelled us to revise our original conclusions for this class of neurons.

In response to your third recommendation, we added a discussion of the limitations of studying sex-determining genes through constitutive mutants and through cell-specific genetic manipulation (see “An organismal sex and a cellular sex” section in Discussion of “Sex-determining genes distinctly regulate courtship capability and target preference via sexually dimorphic neurons”). We believe that cell-specific manipulation of sex-determining genes is a powerful method that is not without weaknesses, and will serve complimentary roles to constitutive mutants when studying their effects on neuroanatomy and behaviors.

Reviewer #1:In both flies and mammals, multiple sexual differentiation mechanisms guide display of adult social behaviors. In flies it is dsx and fru, whereas in mammals it is largely testosterone and estrogen. Ishii et al. present a wide-ranging set of observations that they use to suggest that various aspects of sexually dimorphic courtship and aggression displays in flies are controlled by either dsx or fru. This seems an over-interpretation of the findings. I am supportive of the study but have a few suggestions.1) Data from Figure 3 is used to conclude that dsx specifies both P1 and NP2631 neurons. This is inconsistent with the anatomy shown in figure panels, which shows that both dsx and fru contribute. It is true that cell number is dependent on dsx, but the morphology of the arbors indicates that fru is essential as well: E3/J3/K3 are different than C3/H3/I3. Similarly, data from Figure 6 is used to conclude that fruM but not dsx specifies Tk neurons; comparison of E3 and F3 indicates that dsxM is also important. The authors should quantify arborization rather than rely on gross examination for Figure 3 and Figure 6.

We agree that we should have quantified our findings to draw conclusions about the neuroanatomy of these neurons. Reviewer #2 also raised concerns about our anecdotal statements regarding neuroanatomy. We therefore thoroughly re-analyzed our anatomical data by registering brains to a “template” *Drosophila* brain, and by using 3-dimentional segmentation and volume measurements. Please see Materials and methods section for technical details. In essence, we used a non-rigid transformation technique described by Jefferis et al., 2007, to register individual sample brains to the standard unisex *Drosophila* brain described in Bogovic et al., 2018. This allowed us to compare labeling patterns across multiple brains and sexes in the single reference space.

Immunohistochemical labeling can have technical as well as biological variability. Such inter-sample variability might have been perceived as inter-genotype differences. To better represent labeling patterns that are consistent across samples, we averaged the signal intensity of imaged neurons after registration in the standard brain. The resulting images are shown in 2-dimensional projections as well as in 3-dimensional movies. A similar approach was taken to show the neuroanatomy of various *fru*-expressing neurons (Yu et al., 2010; von Philipsborn et al., 2014), and we feel this is the best approach to concisely summarize our observations from multiple samples.

As a quantification, we compared the volumes of specific neuronal innervations across different genotypes. We used the 3-dimensional rendering program FluoRender to segment structures of interest, and calculated the volumes of each structure.

With these new and rigorous approaches, we are now able to quantitatively support our initial conclusions that (1) NP2631 ∩ dsx^FLP^ neurons are largely specified by *dsx*, and (2) Tk-GAL4^FruM^ neurons are specified by *fru*.

In “Sex-determining genes distinctly regulate courtship capability and preference toward females through sexually dimorphic neurons” Figure 3H-K and Video 3, Video 4 and Video 5, we demonstrate that the overall morphology of NP2631 ∩ dsx^FLP^ neurons in males and fruF males overlap well, and that these neurons in females and fruM females are also similar. We quantified three neuropils that show noticeable sexual dimorphism, and in all cases, volumes of these neuropils in fruF males are statistically indistinguishable from those in males. Volumes in fruM females are comparable to values in females except in one neuropil. We acknowledge that our quantification is based on the entire neural population, and that we may not have sufficient sensitivity to detect either subtype-specific contributions of *fru*, or differences at finer scales such as number or length of branches. We mention this possibility in the main text. Overall, however, our analysis provided us with evidence arguing that sexual dimorphism of NP2631 ∩ dsx^FLP^ neurons is mainly, if not entirely, specified by *dsx*.

Likewise, data in “Layered roles of *fruitless* isoforms in specification and function of male-type aggression-promoting neurons in *Drosophila*” Figure 1L, M and Video 3 clearly show that average Tk-GAL4^FruM^ neurons from males and from fruM females almost perfectly overlap (light cyan in Video 3 indicates the overlap between male (green) and fruM female (dark blue) Tk-GAL4^FruM^ neurons). As is shown in “Layered roles of *fruitless* isoforms in specification and function of male-type aggression-promoting neurons in *Drosophila*” Figure 1N-P, three prominent innervations of these neurons show similar volumes across males (both *fru* +/+ and *fru^M^/fru^4-40^*) and fruM females. We therefore retain our original conclusion that Tk-GAL4^FruM^ neurons are predominantly specified by *fru*.

In contrast, we found that both *dsx* and *fru* are important for specifying P1^a^ neurons. In the previous version, we hastily concluded that *dsx* specified P1^a^ neurons mainly because P1^a^ neurons appear in fruF male brains, but not in fruM female brains. However, quantification of volumes of two neuropils clearly indicates that P1^a^ neurons in fruF males have distinct morphology from those in males (see “Sex-determining genes distinctly regulate courtship capability and preference toward females through sexually dimorphic neurons”, Figure 6F-I). Results from this and re-analysis of behavioral data (discussed below) compelled us to revise our conclusion about P1^a^ neurons. We now fully acknowledge the role of *fru* in both specification and function of P1^a^ neurons. We feel this is a striking contrast to NP2631 ∩ dsx^FLP^ neurons, and further supports the idea that these two neurons are distinct subpopulations within what has been collectively described as P1 or pC1 neurons.

We thank the reviewer for encouraging us to take an extra step for verification (which turned out to require substantial manual and computational works), as we feel our revised conclusions are now better supported by the data.

Presentation of the histology could be better elsewhere too. For example, from panels Figure 2A,C, I conclude that there is no overlap of R15A01 and R71G01, raising the question of the intersectional identification of P1^a^ neurons.

We admit that images in our original version were not sufficiently clear, especially on printed media. It is a challenge to show all visible neurons clearly without over-saturating strongly labeled cells, or without dominating the figure panel. We enlarged the images by 20%, and would like to draw attention to original images provided as source data.

Regarding the overlap of R15A01 and R71G01 promoters, our initial submission used the *R71G01-LexA* transgene in attP40. It is known in the community that transgene expression at the attP40 landing site is not as robust as at the attP2 landing site, which is used for creating the Rubin GAL4 collection. We had created a transgenic animal with *R71G01-LexA* in attP2, and tested its overlap with NP2631 ∩ dsx^FLP^ and with NP2631 ∩ fru^FLP^ (“Sex-determining genes distinctly regulate courtship capability and preference toward females through sexually dimorphic neurons”, Figure 5C, D and Figure 5—figure supplement 3C, D, respectively). As anticipated, we observed more *R71G01-LexA*-labeled neurons from the attP2 transgene than from the attP40 transgene. The overlap with NP2631 ∩ dsx^FLP^ or NP2631 ∩ fru^FLP^ neurons is still very small, confirming our initial conclusion that P1^a^ neurons have minimal overlap with these neurons. We strengthen this conclusion by comparing registered and averaged 3D images of NP2631 ∩ dsx^FLP^ and P1^a^ neurons (“Sex-determining genes distinctly regulate courtship capability and preference toward females through sexually dimorphic neurons”, Figure 5E-G). Neuroanatomy of these two populations shows significant differences.

In addition, Hoopfer et al., 2015 showed that optogenetic stimulation of *R15A01-LexA* neurons recapitulates the behavioural effects of the P1^a^ neuronal activation (see Figure 3H in the citation). This data suggest that *R1501-LexA* indeed includes P1^a^ neurons. We therefore remain confident with our conclusion which this piece of data is most relevant: NP2631 ∩ dsx^FLP^ and P1^a^ neurons are largely separate subpopulations.

We would like to add that split GAL4 lines are often created from lines labeling seemingly different neurons at a population level, especially in z-projection images. In fact, it may be difficult to discern overlapping neurons from the images of R15A01-GAL4 and R71G01-GAL4 in Figure 1C of Hoopfer et al., 2015 (which described P1^a^ neurons).

2) It is clear that FruM is essential for male type lunges toward other males and enhancing wing extension to females. However, FruM is also essential for the pattern of wing extension upon activation of P1^a^ neurons; in controls (Figure 1), wing extensions in phase 2>3 and 2<3 toward males and females, whereas in FruF males (Figure 4A-D), wing extensions in phase 2<3 toward males and females. In addition, it is clear that FruM is essential for regulating intensity of wing extensions. It is a gross simplification to conclude that dsxM enables execution of male courtship whereas FruM enforces choice of target sex. Clearly, FruM is also critical for execution of this behavior.

The reviewer was right in pointing out the importance of *fru* on the function, as well as on neuroanatomy (discussed above), of P1^a^ neurons. We confirmed that optogenetic stimulation of P1^a^ neurons in fruF males induced significantly less wing extensions than in regular males regardless of target sex (“Sex-determining genes distinctly regulate courtship capability and preference toward females through sexually dimorphic neurons”, Figure 6—figure supplement 2). This means that the courtship-executing function, as well as specification, of P1^a^ neurons require the activity of both *dsx* and *fru*. Currently, we do not know whether *dsx* and/or *fru* is necessary within P1^a^ neurons, or whether the proper function of P1^a^ neurons requires another population of *dsx*- or *fru*-dependent neurons (see also discussions regarding cell type-specific manipulation of sex-determining genes). We therefore reasoned that it would be prudent to avoid speculations about the roles of *dsx* and *fru* on P1^a^ neurons, and decided to re-frame P1^a^-related data in the context of discussing the potential diversity of the P1/pC1 neuronal cluster.

On the other hand, optogenetic stimulation of NP2631 ∩ dsx^FLP^ neurons in fruF males induced a similar amount of wing extensions as the same manipulation in regular males did toward male target flies (“Sex-determining genes distinctly regulate courtship capability and preference toward females through sexually dimorphic neurons”, Figure 4C). Together with our observation that the neuroanatomy of NP2631 ∩ dsx^FLP^ neurons in regular males and in fruF males showed little difference, we conclude that (1) fruF males are capable of executing wing extensions, (2) NP2631 ∩ dsx^FLP^ neurons are one neuronal component that belongs to a *dsx*-dependent wing extension execution circuit, and (3) fruM is important for enhancing courtship behavior specifically toward females.

By referring to the “execution” of courtship behavior, we are specifically focusing on the animal’s capability to express courtship behavior. We aim to uncover a neurogenetic process that enables animals to perform courtship behavior at all, in contrast to the ability to adjust the intensity of courtship behavior toward females. Please note that we *do not* argue that *fru* is not required for proper courtship behavior. What we argue is that *fru* is not necessary for *all* aspects of male courtship behavior. The importance of *fru* on male sexual behavior has been extensively demonstrated, leading some to state that fruM specifies all aspects of male sexual behaviors. Here, we wish to shed light on the fundamental characteristics of *fru* mutants, which is the unusually high level of courtship behavior toward males, not the absence of courtship behavior altogether. This phenotype has been consistently and repeatedly reported (literature include Gailey and Hall (1989), Villella et al. (1997), Ito et al. (1996), Demir and Dickson (2005), Shirangi et al. (2006), Pan and Baker (2014), among others), even though these observations may not have received sufficient attention. Moreover, many of previously characterized *fru*-expressing neurons are sensory neurons or early-stage interneurons that transmit sensory information. We feel that our conclusions are entirely consistent with past literature. Our novel functional characterization of NP2631 ∩ dsx^FLP^ neurons provides a neuronal substrate that accounts for the courtship behavior-execution capability of fruF males.

3) Why do males not lunge toward other males in the absence of activation of P1^a^ or NP2631 or Tk neurons (Figure 1, Figure 6)?

The tester flies were all housed in groups for 6 days (P1^a^ neurons) or 14 days (for NP2631 ∩ dsx^FLP^ neurons), which potently reduces spontaneous aggressive behavior. Socially isolated animals show high levels of spontaneous aggression, which can cause a ceiling effect when trying to observe further increases of aggression. Both in flies and mice, group-housing is a common treatment to characterize behavioral changes in response to neuronal manipulations.

4) There is some over-simplification of the literature. It might be true in flies that most studies have only used one sex as a target individual to analyze function of particular neurons (subsection “Effect of a target fly’s sex on optogenetically induced social interactions”), but this is certainly not true in mice (studies from Anderson, Dulac, Lin, Shah).Rather than draw parallels to the four core genotype in the mouse (using which not a single gene has been identified that regulates social behavior independent of sex hormones, subsection “dsx, and not fru, specifies both P1^a^ and NP2631 ∩ dsx^FLP^ neurons”), it would be more appropriate to discuss the role of estrogen vs. testosterone in regulating male behaviors. It is clear that estrogen masculinizes the brain during a critical window whereas testosterone signaling amplifies male behaviors in adult life. This would be a more instructive analogy when trying to dissect the specific functions of dsx and fru.

We appreciate the reviewer’s constructive suggestion to re-frame the roles of *dsx* and *fru*. Indeed, studies in mice often use both males (often castrated) and females as intruders. While that these neuronal manipulations may appear to trigger social behaviors largely independent of target sex, we found that some target sex-dependence has been previously reported. For instance, Lin et al. (2010) reported that optogenetic stimulation of the mediolateral part of the ventromedial hypothalamus (VMHml) triggered aggression more consistently toward males than toward females (Figure 4L,M in the citation). Likewise, close examination of data from Yang et al. (2017), which investigated the behavioral changes caused by chemogenetic activation of progesterone receptor expressing neurons in VMHml, suggests that aggression toward males was more consistent that toward females, although it was not explicitly tested (Figure 1F,G in citation). These past observations reinforce our idea that target sex in neuronal manipulation experiments can be an important variable in other experimental models.

We also agree that the “four core genotypes” model has not been particularly successful in providing mechanical insight into the origin of sexual dimorphism. Although the temporal dynamics between mammalian sex hormones and *Drosophila* sex-determining genes can be quite different, the reviewer’s opinion that the roles of *dsx* and *fru* can be analogous to those of estrogen and testosterone is indeed a quite useful framework for our study. We incorporated this suggestion into our Discussion section.

Reviewer #2:Unfortunately, I have some major technical concerns with the manuscript, and the lack of quantitative anatomical analyses throughout. The rather cursory analysis offered in this manuscript has several critical caveats that are not discussed and undermine the conclusions that can be drawn from the experimental data. Therefore, this manuscript is not suitable for publication without significant experimental clarification of these points.

As discussed in response to reviewer #1, we incorporated quantification of our neuroanatomical data. Additional details are provided below, and we hope these new analyses address the reviewer’s concerns.

Subsection “Effect of a target fly’s sex on optogenetically induced social interactions” and throughout the manuscript:Neither the P1^a^ split-Gal4 nor NP2631 ∩ dsx^FLP^ driver are 'P1/pC1'-specific, some additional neurons in the brain express Gal4 in both cases (see both Hoopfer et al., 2015 and Koganezawa et al., 2016). Therefore, by using these drivers, the authors cannot conclude that the observed behavioral phenotypes were caused purely by the purported 'P1/pC1' neurons. Further intersectional approaches to rigorously target these neurons are essential.

We are a little confused by the reviewer’s assertion that “Neither P1^a^-GAL4 or NP2631 ∩ dsx^FLP^ are ‘P1/pC1’-specific”, after citing the previous publications. In both cases, we see in the citation figures that these drivers in fact label a single cluster of neurons cleanly, both in original publications as well as in our own studies.

For P1^a^-GAL4 neurons, Figure 1E of Hoopfer et al., 2015 shows clearly that only a single cluster of neurons at the posterior end of a male brain were labeled. We confirmed this specificity repeatedly, as shown in “Sex-determining genes distinctly regulate courtship capability and preference toward females through sexually dimorphic neurons”, Figure 6A, Figure 6—figure supplement 1A, C. NP2631 ∩ dsx^FLP^ neurons described in Figure 4D of Koganezawa et al., 2016 also seems to label a well-defined single cluster of neurons per hemisphere. Again, we observed a bilateral single cluster of neurons at the posterior edge of male brains (“Sex-determining genes distinctly regulate courtship capability and preference toward females through sexually dimorphic neurons”, Figure 3C, Figure 3—figure supplement 1A, C). Please also see the movies showing the 3-dimentional structure of P1^a^ and NP2631 ∩ dsx^FLP^ neurons.

We wondered whether the reviewer was raising the possibility that this seemingly single cluster of neurons may indeed consist of heterogeneous populations. Although the number of cells that are labeled by both drivers is rather small (“Sex-determining genes distinctly regulate courtship capability and preference toward females through sexually dimorphic neurons”, Figure 3G and Figure 6E), further intersection could reveal a specific subset that is responsible for the observed behaviors. We currently do not have a means to reliably separate such subpopulations, and we also feel that such experiments would better belong to a future study. Here, we focus on the existence of *dsx*-dependent neurons that can execute wing extensions, and a separate, *fru*-dependent mechanism that enhances wing extensions toward females.

*Subsection “dsx, and not fru, specifies both P1^a^ and NP2631* ∩ dsx^FLP^ neurons”:The expression 'P1^a^ neurons were specified only in the flies that had dsxM' is not accurate and is likely even wrong. Whether the presence of the P1 neurons depends on fru or dsx was originally examined in the very first P1 paper, Kimura et al., 2008. Kimura et al. previously demonstrated by using dsx mutations that the absence of dsxF, but not the presence of dsxM, is the most striking factor for determining the presence of the P1 neurons.In Figure 3, the dsx locus was always wild type and therefore the authors could not dissociate the presence of dsxM and the absence of dsxF. It is essential for the authors to do the similar experiments in combination with the appropriate dsx mutant alleles.

We admit that we were careless in stating that either dsxM or dsxF is responsible for neuroanatomical or behavioral differences from regular males. With the reagents we used, we could only state whether sexually dimorphic splicing of *dsx* has an impact on NP2631 ∩ dsx^FLP^ or P1^a^ neurons. We do not know whether altered neuroanatomy of P1^a^ neurons in fruF males is due to the lack of dsxM, or due to the presence of dsxF. As the reviewer points out, we must use *dsx* mutants (especially dsx^Dom^, which forces male-type splicing of *dsx* regardless of sex chromosome composition) to gain insight into which of the two isoforms is responsible.

However, we would like to emphasize that it is not the specific role of dsxM or dsxF, but rather the genetic origin of sexual dimorphism, that we wished to investigate in this study. That is, we are interested in addressing which sexually dimorphic characteristics (neuroanatomical and behavioral) are controlled by the dimorphism (e.g., sex-specific splicing) of *dsx* and *fru*. As we state in the Discussion, one sex is not a mutant of the other sex, and sex-determining genes often have different functions in each of the two sexes – although *fru* was often considered silent in females, we presented circumstantial evidence that the fruF allele may not be truly null (“Layered roles of *fruitless* isoforms in specification and function of male-type aggression-promoting neurons in *Drosophila*”, Figure 1—figure supplement 3A-C). In this context, splicing mutants such as fru^M^ and fru^F^ provide a useful platform to understand the genetic nature of sex-specific transformation. These two alleles have the added benefit that experiments could be conducted in a symmetrical manner – we can exhaust all four possible combinations of *dsx* and *fru* sex-specific isoforms (“Sex-determining genes distinctly regulate courtship capability and preference toward females through sexually dimorphic neurons”, Figure 3B). Currently, no mutation that forces female-type splicing of *dsx* is known.

We believe our data present strong evidence that *dsx* and *fru* have distinct impacts on the specification and function of sexually dimorphic neuronal circuits. We revised our statements and referred to the importance of *dsx* as a gene, not of sex-specific splicing isoforms.

The authors mentioned that 'processes of P1^a^ neurons in fruF males appeared thinner than those in males'. What does thinner mean? What form of scientific measurement is "thinner"? This needs to be quantified properly. Image co-registration or any other quantitative analyses for the size differences of the neuronal processes between these genotypes must be carried out.The authors also mention ‘sex differences both in terms of cell body numbers and branching patterns’. If the authors declare the branching patterns were different between genotypes, they must carry out quantitative analyses.They also declare the neuronal morphology was 'similar' between particular genotypes, but what does 'similar' mean? The authors seem to conclude that the neuronal morphology is different when they expect it is different and that it is similar when they expect it is similar, without any quantitative criteria for measurement.

As stated above in reply to a similar request from reviewer #1, we did an extensive re-analysis of our neuroanatomical data. We followed reviewer #2’s request, and hope that our new set of data will directly address the reviewers’ concerns.

Subsection “P1^a^ and NP2631 ∩ dsx^FLP^ neurons in fruF males can induce courtship, but not male-type aggressive behaviour”:The logical link between the anatomy in Figure 3 and behavior in Figure 4 is unclear. In Figure 3, the authors looked at the contributions of the fru and dsx genes in the purported 'P1/pC1'-specific neurons, but in Figure 4, they carried out behavioral assays with the fru mutants, in which fru gene function is disrupted not only in P1/pC1 neurons but also all other fruM-expressing neurons throughout the PNS and CNS, they then base all their conclusions on only the behavioral function of P1/pC1 neurons. These animals will have a plethora of fru mutant phenotypes that are associated with lack of fruM in other neurons in the CNS and PNS.To make any relevant conclusions the authors need to re-do the behavioral assays in Figure 4 using the flies in which only the P1/pC1 neurons are mutant for fru (e.g., mutant MARCM clones, UAS-Cas9 system, or fru-RNAi).

Our experiments address what the NP2631 ∩ dsx^FLP^ or P1^a^ neurons can do in fruF males. We would like to clarify that our original submission *did not* state that *fru* is required cell autonomously in either NP2631 ∩ dsx^FLP^ or P1^a^ neurons, precisely because we were aware that we could not attribute the behavioral differences from regular males to the deficit specifically within either neuronal population. We apologize if our wording in the previous submission caused any confusion.

We agree with the reviewer’s critique that the use of constitutive mutants has a limitation. We cannot answer whether the phenotype (anatomical or behavioral) is due to the lack of *dsx* or *fru* functions in the neurons of interest, or due to their cell non-autonomous functions (likely in other neurons). However, we would like to reiterate that the cell-autonomous function of *dsx* or *fru* is not the focus of this study. We wished to address how sexual dimorphism of *dsx* and *fru* (in a form of sex-specific splicing) may contribute to the specification of sexually dimorphic circuits and sexually dimorphic social behaviors. As stated above, simple loss-of-function of *dsx* or *fru* does not necessarily distinguish which male-like or female-like characteristics are specified by a given gene, because the loss of a gene is not synonymous to the transformation of sex from one to the other. We respectfully disagree with the reviewer’s assertion that we cannot make “any relevant conclusions” with these mutants. Rather, we believe that systematic investigation into what changes in fruF males and fruM females at circuit and behavioral levels (e.g., comparison of all 4 genotypes shown in “Sex-determining genes distinctly regulate courtship capability and preference toward females through sexually dimorphic neurons”, Figure 3B) is crucial for understanding the transformative nature of sex determination.

We would also like to be mindful of the limitations of cell-type specific manipulations of *dsx* and *fru*. For instance, successful gene knockdown using ‘dead Cas9’ (CRISPRi) crucially depends on the sequence of 20-base guide RNA. RNA interference (RNAi) relies on the specificity of ~21-base double-strand RNA sequence. In both cases, off-target effects as well as incomplete (or even ineffective) knockdown of the target gene are prevalent and well documented. Incomplete knockdown can be also caused by varied strengths of GAL4 drivers. To make matters more complicated, *dsx* and *fru* are likely crucial during development. This means that a GAL4 driver needs to turn on very early and remain active consistently during development to provide interpretable results. In previous studies, characterization of both the efficacy of *fru* RNAi knockdown or the temporal activity or GAL4 drivers is disturbingly sparse, raising a question about the interpretability of such data. Presumably because of this limitation, results of RNAi against sex-determining genes are often presented only when such manipulation results in reproduction of phenotypes in systemic mutants (that is, when the result is interpretable). Considering that cell-autonomous role of *dsx* or *fru* is not necessarily the focus of our study, we do not think either RNAi or CRISPRi will further support our conclusions.

MARCM can circumvent the problem of timing and strength because the mutations become homozygous at the cell division. However, MARCM also has an often-overlooked weakness that any cell clones in which a GAL4 driver is not active remain undetected. Moreover, creation of animals with mutant cells only in the desired neuronal population is exceedingly difficult, often requiring hundreds of individuals to identify a handful of suitable samples. Because the distribution of cell clones can be analyzed only after behavioral experiments, and since there is substantial inherent variability of behavior, an unreasonable amount of labor would be required to obtain interpretable data using MARCM.

We are aware of the limitation of constitutive mutants, but systematic analysis of such mutants is fundamental to interpret cell-specific manipulations of sex-determining genes. We hope reviewers will understand the value in what we can observe, and allow us to address functions of *dsx* or *fru* within specific neuronal populations in future studies.

Subsection “fruM specifies aggression-promoting Tk-GAL4^FruM^ neurons”:The logic of transition from P1/pC1 to Tk-Gal4 neurons is very difficult to follow. The relationship between the former and the latter need to be explained. Are they completely different stories?

As mentioned at the beginning of this letter, we have split the original submission into 2 parts. This point was also raised by reviewer #3, and we hope the current format better conveys our central messages.

Figure 6:As a control, the authors should demonstrate that Tk-GAL4 neurons do not express dsx.

Please see Figure 1 —figure supplement 2C.

Subsection “Differential roles of fru isoforms on male-male interactions and specification of Tk-GAL4^FruM^ neurons”:The authors showed that the Tk-Gal4 neurons exist in fru^∆B^ mutant males but behavior is affected in these flies. The simplest explanation would be that the neurites are malformed in these flies.Although the authors mentioned that 'fruMA and fruM mutants did not affect […] their arborization pattern', they didn't seem to carry out any quantitative morphological analyses. Such analyses should be included.Subsection “The Tk-GAL4^FruM^ neurons in fruMB mutants can induce male aggression”:The authors mention that 'these mutants have Tk-GAL4^FruM^ neurons that appear to retain their morphology'. But how did they examine this?

We now show the averaged images of Tk-GAL4^FruM^ neurons in fruMA and fruMB mutants, which is in fact very similar to those in wild type males (“Layered roles of *fruitless* isoforms in specification and function of male-type aggression-promoting neurons in *Drosophila*”, Figure 5F-H, Video 7). The volume quantification of three major neural processes in these three genotypes indeed shows that they are indistinguishable from each other.

Again, this level of analysis may not reveal subcellular differences in neuromorphology among regular males, fruMA mutants, and fruMB mutants. However, we would like to point out that our conclusion that fruMB does not affect the specification of Tk-GAL4^FruM^ neurons is based on physiological and behavioral data, as well as neuroanatomical data (“Layered roles of *fruitless* isoforms in specification and function of male-type aggression-promoting neurons in *Drosophila*”, Figure 6 and Figure 5, respectively). The fact that optogenetic stimulation of Tk-GAL4^FruM^ neurons in fruMB mutants triggers aggression as robustly as in regular males suggests that the overall integrity of Tk-GAL4^FruM^ neurons remains largely unaffected in fruMB mutants.

Reviewer #3:This paper is a paradox. As far as I can tell it is rigorous on all levels. The scholarship is excellent, deep and well rounded. In addition, the issues are explored thoroughly and there is an enormous amount of data. In addition, an enormous number of fly lines (the table that describes the lines is phenomenal). But on the other hand, it is too much. I would like to discuss the argument and conclusions that surface relating dsx and fru expressing neurons with sexually dimorphic choice vs. sexually dimorphic motor patterns and the interesting differences between reproductive behavior and aggression, but the paper is simply too dense at almost every point. Reading it is a painful experience and this does the work a tremendous disservice. I say this recognizing and admiring the care that has gone into preparing this text. It seems flawless, just impossibly dense.

We understand a reservation that echoes with other reviewers’ concerns regarding the clarity of our original submission. We would like to address the reviewer’s specific suggestions in the following sections.

What to do? Here is a list of suggestions.1) The text and figures are impeccably constructed but for the average reader they are overly complicated and far too dense. The flow of the manuscript would benefit greatly from severe editing of the text, figures and figure legends. It is with a heavy heart that I ask the authors to do this knowing the amount of effort it took to construct the manuscript.

We significantly reorganized our original submission. Namely, we first understood the value of the last suggestion and split the original version into two new manuscripts as we already described. We hope this will substantially improve the clarity of our messages.

2) Remove raster plots from figures throughout and move to supplemental material if needed. See Figure 1, Figure 4, Figure 5, Figure 6, Figure 7, Figure 9.

We moved the raster plots of behaviors largely to figure supplement panels.

3) Figure 2 is not required. Move to supplemental material and concisely state result in text.

We feel the evidence suggesting that P1^a^ and NP2631 ∩ dsx^FLP^ neurons are separate populations has significance, especially now that we detected different levels of contribution of *fru* on these two neuronal populations. We therefore decided to keep the panels in “Sex-determining genes distinctly regulate courtship capability and preference toward females through sexually dimorphic neurons”, Figure 5.

4) Much of the immunohistochemistry could be removed from figures and moved to supplemental material. See Figure 3, Figure 6, Figure 8.

This is in contrast to requirements from reviewers #1 and #2 to perform more rigorous neuroanatomical analyses. We followed a custom for most *Drosophila* papers and kept representative images, but we would be open to further suggestions for improving clarity.

5) The summary panels at the end of some of the figures are helpful. But a single unifying summary/model would be best. Though organized in a systematic and rational way the results still feel piecemeal and it is tiresome to continually have refer back to previous figures or supplemental material to regain your bearing. A single model would help immensely.

We reduced the number of models to one for “Sex-determining genes distinctly regulate courtship capability and preference toward females through sexually dimorphic neurons” (Figure 4D), and two for “Layered roles of *fruitless* isoforms in specification and function of male-type aggression-promoting neurons in *Drosophila*” (Figure 2D and Figure 6H). It was a challenge to summarize our findings in a single model, given that *dsx* and *fru* have different roles on 3 populations of neurons and 2 types of social behaviors. We hope the new models make our core finding clear.

[Editors’ note: what follows is the authors’ response to the second round of review.]

Essential revisions:Despite the notable revisions, e.g. valuable inclusion of quantitative anatomical analyses of the P1/pC1 neurons, there are remain issues that remain to be addressed.

We appreciate the reviewers’ comments on key issues. We would like to respond to each critique below.

1) Although the cover letter refers to Figure 1E of Hoopfer et al. (2015) to justify the view that the driver is P1/pC1-specific, Figure 1—figure supplement 1C of Hoopfer et al. (2015) appears to show that there are several additional neurons close to the lateral horn labelled by this split-Gal4 driver in both males and females. The main text of Hoopfer et al. (2015) states that "in addition to P1 neurons, the P1^a^ spGAL4 driver labelled a minor and variable population of neurons (<4 per hemi- brain) with cell bodies in the lateral horn (LH) that were neither FruM-positive nor male-specific".2) The same issue applies to the NP2631 ∩ dsx^FLP^ driver. The text and cover letter reference Figure 4D of Koganezawa et al. (2016) to justify that their driver is P1/pC1-specific. However, the referenced figure appears to show a single descending axon per hemisphere is going out of the ventral end of the GNG. This axon crosses the midline at the neuropil region just dorsal to the oesophageal foramen, which is a typical neurite structure of another dsx-expressing neuronal cluster called pMN1 (see Figure S1 of Kimura et al., 2015). Similarly, in Figure S3A of Koganezawa et al. (2016), a descending axon of pMN1 is more obvious in this female specimen (pMN1 is present in both sexes with dimorphic neurite morphology, according to Figure S1 of Kimura et al., 2015). I would encourage the authors to show the expression patterns in the VNC. Are there any cells in this tissue labelled by the NP2631 ∩ dsx^FLP^ driver?The above two Points 1 and 2 reflect concerns on the specificity of driver lines used. To address ambiguities that arise from this, unless there are additional strong arguments in support of the authors' interpretations, ideally all behavioural assays should be re-done under the conditions of P1/pC1-specific activation using either further intersection or a stochastic/mosaic approach.

We certainly share the concern over the specificity of GAL4 drivers. When we attempted to make a causal link between the manipulated neurons and observed behavioral phenotypes, it is critical to know in which cells the proteins that manipulate neural activities (effector proteins) are present. This is why we chose to directly visualize the protein (CsChrimson:tdTomato) that serves as the effector of neuronal manipulation. As we wrote in our main text in “Sex-determining genes distinctly regulate courtship capability and preference toward females through sexually dimorphic neurons”, subsection “The target fly’s sex affects the function of social behavior-promoting neurons”, this should eliminate the necessity to deduce the expression pattern of the effector protein that cannot be visualized based on expression pattern of a reporter protein from other UAS elements. The same GAL4 line shows a dramatically different expression pattern when combined with different UAS elements (see Pfeiffer et al., 2010 and 2012 for examples), requiring caution when extrapolating the expression of reporter proteins (GFP for both Hoopfer et al., 2015: Figure 1—figure supplement 1C, and Koganezawa et al., 2016: Figure 4D) to deduce expression of untagged, and therefore invisible, effector proteins (dTRPA1). Such interpretation is widely accepted in numerous publications.

Keeping this in mind, we first wish to reiterate that we did not detect any cell bodies other than P1^a^ neurons when examining the expression pattern of CsChrimson:tdTomato driven by the P1^a^ split GAL4 from any brain (Figure 6A-G and Figure 5—figure supplement 1C, D).

In contrast, we occasionally detected a CsChrimson:tdTomato-expressing neuron outside of P1/pC1 cluster of NP2631 ∩ dsx^FLP^ neurons (which in contrast were found in all male brains examined: shown in Figure 3C-K, Figure 2—figure supplement 2A_1_). We apologize for not mentioning the presence of this neuron in our initial submission. As shown in the new Figure 2—figure supplement 2A_2_, this neuron projects contralaterally and to the VNC. We appreciate the reviewer’s eye to identify this neuron in Figure 4D of Koganezawa et al., 2016, the image quality of which was too poor for us to discern a descending projection from noise, and to characterize this as “pMN1” neuron described in Supplementary Figure S1 of Kimura et al., 2015, without an image from VNC. In our samples, this descending neuron seems to have dense arborization in the ventromedial neuropil (nomenclature based on Ito et al., 2016), which is missing in examples shown in the above-mentioned Kimura et al., 2015.

However, the presence of this descending neuron is not consistent enough to account for behaviors induced by the optogenetic stimulation of NP2631 ∩ dsx^FLP^ neurons. The descending neuron is labeled only in 35% of the male brains analyzed in Figure 2—figure supplement 2D_1_. These ratios are significantly lower than the ratio of the flies that showed LED stimulation-dependent lunges (Figure 2—figure supplement D_2_) or wing extensions (Figure 2—figure supplement 2D_3_). While the contribution of the descending neuron on social behaviors cannot be excluded, these observations strongly argue that they cannot account for the behavioral effect that we observed as a result of the stimulation of NP2631 ∩ dsx^FLP^ neurons.

We therefore conclude that we found little evidence that re-doing behavioral experiments using “further intersection” will change our conclusions. While we agree that additional neurons may be labeled by using different UAS reporter elements, the combination of the GAL4 driver and UAS effector we used in this manuscript provides sufficient specificity to maintain that “P1/pC1” clusters labeled by NP2631 and dsx^FLP^ are causally responsible for the behaviors we observed. Of course, the specificity of observed expression depends on other technical limitations such as the antibody sensitivity, detection threshold of the microscope, and so on, but these limitations also apply to other published works.

We added a paragraph that summarizes our statements above in “Sex-determining genes distinctly regulate courtship capability and preference toward females through sexually dimorphic neurons”, subsection “NP2631 ∩ dsx^FLP^ neurons contain subpopulations of the P1/pC1 cluster that promote both courtship and aggression”. Also, we speculate that both (P1/pC1 cluster of) NP2631 ∩ dsx^FLP^ neurons and P1^a^ neurons can be further subdivided into distinct neuronal classes, as we mentioned in our discussion (“Sex-determining genes distinctly regulate courtship capability and preference toward females through sexually dimorphic neurons”, subsection “Uncovering functional heterogeneity of social behavior-controlling neurons”).

3) It still remains unclear how fru function is dissociated from that of dsx in the sexual differentiation of P1/pC1 neurons. To clarify potentially distinct contributions of fru and dsx to this developmental process, it appears that dsx mutant analyses are required. Have these been attempted or done?

We are a little perplexed by this comment. As we explained in the schematics of Figure 3B, our 4 genotypes allow us to dissociate the contribution of *dsx* isoforms and *fru* isoforms on circuit formation and behavior. For instance, if the male-like phenotype appears in the dsxM/fruF genotype, but not in the dsxF/fruM genotype (as is the case for NP2631 ∩ dsx^FLP^ neuronal morphology), we can certainly conclude that it is the *dsx* isoform, and not the *fru* isoform, that controls the phenotype. We are not sure exactly what alternative possibilities the reviewers think there would be, and how additional experiments with *dsx* mutants would help differentiate such possibilities. While *dsx* mutants (especially the one that forces specific isoform splicing, as described in Nagoshi and Baker, 1990) will likely provide additional evidence, we do not think that such experiments are necessary for our conclusions.

Note that we did not attempt to investigate whether it is the lack of dsxM, or the presence of dsxF, that is responsible for the phenotype. We acknowledge that we cannot differentiate these two possibilities using our 4 genotypes (see “Sex-determining genes distinctly regulate courtship capability and preference toward females through sexually dimorphic neurons”, subsection “An organismal sex and a cellular sex” and “Layered roles of *fruitless* isoforms in specification and function of male-type aggression-promoting neurons in *Drosophila*”, subsection “*fru* specifies a neural circuit for sexually dimorphic aggressive actions”), but it is not the question we tried to answer.

4) The experiments use fruM, fruF and single-isoform fru mutant alleles for behavior and attribute the behavioural phenotypes to P1/pC1 (this paper) and Tk-GAL4^FruM^ neurons (the other paper, titled “Layered roles of fruitless isoforms in specification and function of male aggression-promoting neurons in Drosophila”). Given that ru isoforms are expressed in other neurons in the CNS and the PNS, surely the effects of Fru isoform on behavior in P1/pC1 and Tk-GAL4^FruM^ neurons can only really be assessed by looking at animals were the specific neurons are mutant for fru (e.g. fru isoform-specific mutant MARCM clones, UAS-Cas9 system, or fru-RNAi)? Otherwise, the data would need to be interpreted with obvious caveats.

As we had acknowledged in the “An organismal sex and a cellular sex” section of the Discussion of “Sex-determining genes distinctly regulate courtship capability and preference toward females through sexually dimorphic neurons”, we had been aware that our experiment does not clarify whether *dsx* and *fru* acts cell-autonomously or through other parts of the nervous system. While the molecular mechanism by which *dsx* and *fru* specify neuronal sexual dimorphism is an important question, we would also like to clarify that “the effects of Fru isoform on behavior in P1/pC1 and Tk-GAL4^FruM^ neurons (in the other manuscript, titled “Layered roles of *fruitless* isoforms in specification and function of male aggression-promoting neurons in *Drosophila”*)” is not the question we aimed to answer.

Based on the fact that *dsx* and *fru* are both transcription factors, a straightforward prediction is that these genes work cell-autonomously, but it is certainly possible that *dsx* and *fru* in other neurons direct the morphogenesis or behavioral impact of P1^a^ and NP2631 ∩ dsx^FLP^ neurons. Such a cell non-autonomous role for *dsx* has been previously reported in the gonad and the genital imaginal disc, but we are not aware of any published results suggesting that *dsx* or *fru* functions cell non-autonomously in the nervous system. In fact, one of the reasons why we did not address this question is because none of the approaches the reviewer suggested (RNAi, dCas9, MARCM) can immediately differentiate these two possibilities. We stated this limitation in subsection “An organismal sex and a cellular sex” (“Sex-determining genes distinctly regulate courtship capability and preference toward females through sexually dimorphic neurons”), and will clarify our arguments below.

An overlooked limitation of RNAi, dCas9, or MARCM approaches is that results are immediately interpretable *only if* it recapitulates the phenotype of the systemic mutant. To further elaborate:

1) Both RNAi and dCas9 are limited by the temporal as well as spatial expression of the GAL4 driver. This means that a GAL4 driver that turns on late may not be effective in suppressing gene activity at the critical developmental phase. For instance, *dsx*-dependent sexual dimorphism of leg gustatory receptor neurons begins to be established as early as 8 hours after pupal formation (Mellert, Robinett, and Baker, 2012). This type of situation leaves open the possibility that knock-down of *fru* using *P1^a^-GAL4* or *NP2631* may occur too late to impact the fate of the neurons. We do not say it *is* too late: we merely say that we cannot interpret the “no phenotype” result (which would be the case when the RNAi turns on too late to impact the cell-autonomous role of *fru*, and when *fru* is indeed required in non-cell-autonomous manner) unless we characterize the expression patterns of a given GAL4 used throughout development. A GAL4 line that fulfills temporal as well as spatial specificity for the targeted cells will be exceedingly difficult to identify.

2) Another seldom verified limitation of RNAi and dCas9 is its efficacy. Both approaches rely on the specificity of 17-23 base homology of RNA substrates, and how efficiently the double-stranded RNA can recruit effector proteins (RISC complex for RNAi, Cas9 for dCas9). Most likely, they provide only partial knockdown of the targeted gene. As we showed in “Layered roles of *fruitless* isoforms in specification and function of male-type aggression-promoting neurons in *Drosophila*”, a hypomorph of *fru* can result in an intermediate phenotype (see Figure 1—figure supplement 3E of that submission). Such incomplete phenotypes can complicate the interpretation when the phenotypic differences of systemic mutants are quantitative rather than qualitative, as is the case for NP2631 ∩ dsx^FLP^ neurons (Figure 3G-N) and P1^a^ neurons (Figure 6E-I)).

3) A limitation of MARCM that evades scrutiny is that the mutant cell clones may be generated in *GAL4 negative cells* as well as GAL4 positive cells. Only GAL4 positive cells are visible, but this does not mean that they are the only mutant cells. This leaves a possibility that a heat-shock condition that generates NP2631 ∩ dsx^FLP^ or P1^a^ neuronal clones may systematically generates other clones together, due to similar developmental time course (note that MARCM works during cell division). This means that MARCM data is inherently correlational, not causal. In fact, a well-cited paper using MARCM for this type of analysis (Kimura et al., 2008) does not exclude the possibility that behaviorally relevant neurons are actually *fru^NP21^*-negative neurons (such as *dsx*+/*fru*- subclass within P1/pC1 neurons) that become *tra* mutant clones together with the *fru^NP21^*-expressing “P1” neurons. Considering the shared lineage of this cluster (the DM4 lineage described in Ren et al., 2016), this is an alternative scenario that remains to be addressed.

4) The other practical limitation of the MARCM approach is the difficulty of characterizing labeled cell types, especially in the P1/pC1 cluster. Our data (Figure 5, Figure 5—figure supplement 1) as well as previous publications (Costa et al., 2016; Zhang et al., 2018) point to the heterogeneity of this neuronal population (discussed in “Uncovering functional heterogeneity of social behavior-controlling neurons”). Morphology of these neurons are often very similar to each other, and it will be extremely challenging to accurately classify labeled neurons especially if multiple types are simultaneously labeled.

These issues can be properly addressed with adequately validated tools (such as truly effective RNAi or gRNA, GAL4 lines with verified temporal and spatial expression patterns, etc.). However, we think conclusive evidence will require a level of effort that amounts to a separate project. In fact, we think it is highly valuable to characterize the neuronal morphology and function in the systemic mutants as we did in our manuscript, precisely because this will be an important foundation on which we can further investigate the molecular mechanism underlying the roles of *dsx* and *fru* on sexual dimorphisms (“Sex-determining genes distinctly regulate courtship capability and preference toward females through sexually dimorphic neurons”, subsection “An organismal sex and a cellular sex”). We hope reviewers will understand both the merit of our current study, as well as the challenges to obtain definitive conclusions regarding the cell-autonomy of *dsx* and *fru*.

5) The paper should include better images of Dsx immunolabeling (Figure 2—figure supplement 2E) as the current one is of very poor quality.

We performed additional immunohistochemistry against DsxM and FruM using a new batch of antibodies, which gave us superior signal to noise. We thank Dr. Michael Perry of UCSD for providing us with these antibodies. The newly captured images clearly show that almost all of NP2631 ∩ dsx^FLP^ neurons (Figure 2—figure supplement 2B, C), and all P1^a^ neurons (Figure 5—figure supplement 1C, D), express DsxM. We updated the figure panels and texts accordingly.